# Origins of direction selectivity in the primate retina

Yeon Jin Kim [1], Beth B. Peterson[1], Joanna D. Crook[1], Hannah R. Joo[1], Jiajia Wu [2], Christian Puller [1], Farrel R. Robinson [1,3], Paul D. Gamlin [4], King-Wai Yau[5], Felix Viana [6], John B. Troy [2], Robert G. Smith [7], Orin S. Packer [1], Peter B. Detwiler[8] & Dennis M. Dacey [1,3 ✉]

From mouse to primate, there is a striking discontinuity in our current understanding of the neural coding of motion direction. In non-primate mammals, directionally selective cell types and circuits are a signature feature of the retina, situated at the earliest stage of the visual process. In primates, by contrast, direction selectivity is a hallmark of motion processing areas in visual cortex, but has not been found in the retina, despite significant effort. Here we combined functional recordings of light-evoked responses and connectomic reconstruction to identify diverse direction-selective cell types in the macaque monkey retina with distinctive physiological properties and synaptic motifs. This circuitry includes an ON-OFF ganglion cell type, a spiking, ON-OFF polyaxonal amacrine cell and the starburst amacrine cell, all of which show direction selectivity. Moreover, we discovered that macaque starburst cells possess a strong, non-GABAergic, antagonistic surround mediated by input from excitatory bipolar cells that is critical for the generation of radial motion sensitivity in these cells. Our findings open a door to investigation of a precortical circuitry that computes motion direction in the primate visual system.

[1] Department of Biological Structure, University of Washington, Seattle, WA 98195, USA. [2] Department of Biomedical Engineering, Northwestern University, Evanston, IL 60208, USA. [3] Washington National Primate Research Center, Seattle, WA 98195, USA. [4] Department of Ophthalmology and Vision Sciences, University of Alabama at Birmingham, Birmingham, AL 35294-4390, USA. [5] Departments of Neuroscience and Ophthalmology, Johns Hopkins University School of Medicine, Baltimore, MD 21205-2185, USA. [6] Institute of Neuroscience, UMH-CSIC, San Juan de Alicante 03550, Spain. [7] Department of Neuroscience, University of Pennsylvania, Philadelphia, PA 19104, USA. [8] Department of Physiology and Biophysics, University of Washington, Seattle, WA 98195, USA. ✉email: dmd@uw.edu

About 60 years ago, neurons in the mammalian nervous system sensitive to the direction of visual motion were described in the primary visual cortex of the cat[1,2] and in the retina of the rabbit[3,4]. Thereafter, investigation of the circuit origins and the neural mechanisms for direction selectivity as well as its functional role in motion processing have proceeded on two largely parallel tracks, focusing on diverse motion-processing areas in the primate visual cortex[5] versus the diverse cell types and circuits in the non-primate mammalian retina[6,7]. One reason for this dichotomy is that, while a significant fraction of retinal ganglion cells in non-primate mammals show direction selectivity[8,9], this fundamental neural signal has not been observed in the primate retina, despite dedicated effort[10–12]. Moreover, the apparent lack of significant direction selectivity recorded from neurons at the level of the lateral geniculate nucleus, which relays signals from retina to cortex, has suggested that this property must be encephalized in primates, arising initially from neural circuits within primary visual cortex[13–17].

The most intensively investigated substrate for direction selectivity is the synaptic connection from starburst amacrine cells to ON–OFF direction-selective ganglion cells (ON–OFF DSGCs) in the mouse and rabbit retina. The ON–OFF DSGCs have characteristic bistratified dendrites that tend to recurve back toward the cell body, and are present as four distinct populations that differ in their direction preference[8,18,19]. Direction selectivity originates in the distinctive radial dendrites of a single type of retinal interneuron, the starburst amacrine cell[20,21], which uniquely co-expresses the inhibitory transmitter GABA and the excitatory transmitter acetylcholine[22,23]. Strong evidence indicates that selective, spatially asymmetric inhibitory synapses from starburst dendrites to each of the four ON–OFF DSGC types is the basis for direction selectivity in these ganglion cells[19,24]. In the primate, ganglion cells with dendritic morphology similar to that of ON–OFF DSGCs have been observed previously[25–27]. Moreover, starburst amacrine cells are also present in the primate retina[28,29], and the starburst and putative DSGC dendrites appear to costratify in the inner plexiform layer (IPL)[27,30]. This raises the possibility that direction selectivity is present in the primate retina and perhaps also computed by circuitry comparable to that in non-primate mammals.

Here, we used a distinctive in vitro preparation of the macaque monkey retina in which the photoreceptor-supporting layers—choroid and retinal pigment epithelium—remain attached to the neural retina, permitting adaptation to a high photopic background that elicits a pure cone photoreceptor-driven response to light[31]. A unique in vitro photodynamic staining method then permitted a near-complete classification of ganglion cells and the identification of a morphologically distinct, ON–OFF direction-selective ganglion cell type that is coupled via gap junctions to a polyaxonal, ON–OFF spiking amacrine-cell type that is itself direction-selective. For the first time in the primate, we were able to target starburst amacrine cells, characterizing their light response, receptive-field structure, direction selectivity, and connectome. We found a strong antagonistic surround, arising presynaptically from diverse excitatory bipolar-cell inputs that is critical for the characteristic directionally selective response evoked in macaque starburst cells by radially moving stimuli.

## Results

### Classification of ganglion cell types and identification of a candidate ON–OFF direction-selective cell population.

Parallel visual pathways originate in morphologically and physiologically diverse retinal ganglion cell populations[32–34]. In the mouse retina, over 40 functionally distinct ganglion cell types are present[35–38], with a substantial fraction of them showing direction selectivity[8].

In the primate retina, by contrast, many fewer ganglion cell types have been identified[26,39–45]. A question therefore is whether the account of ganglion cell types in the primate is largely incomplete or whether there is a large species difference in this fundamental aspect of retinal organization. One explanation may be that the abundance of diverse direction-selective types found in non-primate mammals may be absent in the primate. Thus, to identify and characterize a candidate DSGCs in the macaque retina a more comprehensive accounting of the dendritic morphologies and spatial densities of the diverse ganglion cell types was required.

To address this problem in the primate—where gene-based cellular reporters are lacking—we used retrograde photodynamic staining[25]. Injections of the tracer rhodamine dextran were made into the major targets of retinal output: the lateral geniculate nucleus (LGN)[25,46,47], the superior colliculus (SC)[48] and nuclei within the pretectal complex[46]. After retrograde transport, the retina was dissected and maintained in vitro. Exposure of labeled ganglion cells to high photopic light intensities liberated tracer sequestered in the cell body, which then rapidly diffused throughout the cell to reveal their complete dendritic morphology. By this method, we determined for each ganglion cell population its relative spatial density and fundamental mosaic organization. The spatial densities of 19 morphologically distinct ganglion cell types, determined from the regular overlap of neighboring dendritic trees, accounted for ~97% of the total ganglion cell population in the mid-peripheral retina (Fig. 1; Supplementary Fig. 1; Supplementary Table 1). Only five types (ON- and OFF-midget, ON- and OFF-parasol, and the small bistratified cell; Supplementary Fig. 1) were present at a relatively high density, accounting for the great majority of ganglion cells (~80%) that project to the LGN (parasol cells also project to the SC[48]). The remaining ~20% of the ganglion cell population comprised a larger number of distinct cell types, each present at relatively low densities (0.5–1.5%) (Supplementary Table 1). One of these low-density types showed recursively branching dendrites that were bistratified at approximately the same depth as the outer and inner cholinergic bands (Fig. 2d), and in this regard resembled the ON–OFF DSGCs of non-primate mammals[7]. These cells were retrogradely labeled from tracer injections made into either the LGN or the SC; and were not observed after injections into the pretectum (pretectal olivary nucleus, nucleus of the optic tract, and dorsal terminal nucleus of the accessory optic system; see methods for details) but we cannot exclude the possibility of additional projections to these targets. These recursive bistratified ganglion cells tiled the retina uniformly with minimal dendritic overlap (Fig. 2a). Thus unlike the non-primate mammal, where four ON–OFF DSGC types with distinct preferred directions form overlapping mosaics and together made up ~12% of the total number of ganglion cells[49], we observed only a single mosaic that comprised ~1.5% of the total ganglion cells in the peripheral retina (Fig. 2 and Supplementary Table 1). We also observed a second class of recursively branching ganglion cell present in the same retrogradely labeled retinas (the recursive monostratified cell; Fig. 1; Supplementary Fig. 1) that we have not characterized functionally. These cells showed extensively overlapping and fasciculated dendrites (Supplementary Fig. 2) in the ON layer of the IPL and appeared to comprise at least three separate ganglion cell populations with similar morphology.

### The recursive bistratified ganglion cell type is direction selective.

A major advantage of in vitro photostaining was that we could target specific morphologically identified types for physiological characterization[46,47] in the in vitro retina[25]. The

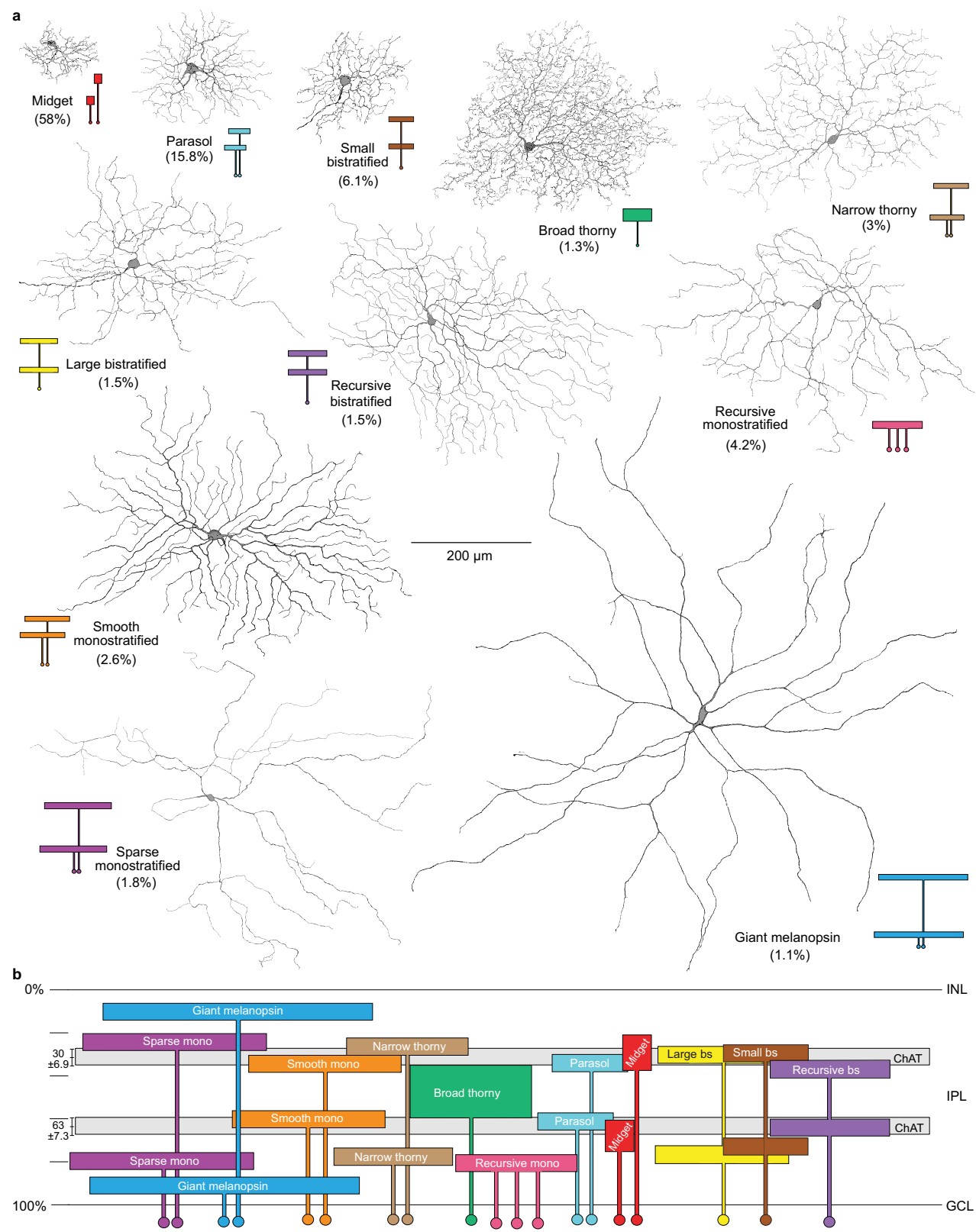

recursive bistratified cells transiently depolarized and spiked at both light ONset and OFFset (Fig. 3b) as expected from their depth of stratification in the ON and OFF subdivisions of the inner plexiform layer (IPL). Direction selectivity was found in response to moving bar stimuli (Fig. 3c, d) ($n = 26$ cells; mean ± s.d., DSI = $0.82 ± 0.20$). The relatively low density of the recursive bistratified cells, and their presence as only a single ganglion cell

type, likely contributed to their apparent absence in early surveys of primate ganglion cell physiology[50]. The preferred direction was variable in this relatively small sample and foveal location was only roughly estimated, however, cells recorded in nearby locations from the same retina tended to show very similar direction preferences (Fig. 3d). We also note that the sharpness of DS tuning but not the preferred direction varied with the parameter

**Fig. 1 Summary of the morphology, stratification and relative spatial densities 19 ganglion cell populations in macaque retina. a** Macaque ganglion cells divided into 11 distinctive morphological groups by dendritic structure, dendritic tree diameter, and mosaic tiling (see Supplementary Fig. 1 for details). These morphological groups bear names that arose historically (midget, parasol) or more recently, and related to specific morphological features of newly identified types (e.g., broad thorny ganglion cells show a unique broad dendritic stratification and fine, thorn-covered dendrites. The color inserts show schematically the cell body (small circle) and dendritic arbor of each cell type indicating how some groups (e.g., midget and parasol) have been further subdivided into types based on stratification within the inner plexiform layer (IPL). Percentages indicate estimated % of total ganglion cell density for that group. The number of types within each named morphological group are indicated by the number of cell bodies associated either with dendrites that stratify at different depths in the IPL (or in the single case of the recursive monostratified cells the same depth). This results in a total of 19 ganglion cell populations that together comprise ~97% of the total ganglion cell population in the peripheral retina (see also Supplementary Fig. 1 and Supplementary Table 1). **b** Stratification depth in the IPL summarized schematically for all cell types in relationship to the outer and inner choline acetyltransferase (Chat) immunolabeled strata formed by the dendrites of starburst amacrine cells (gray bands) as indicated (see Supplementary Figs. 1 and 3 for details).

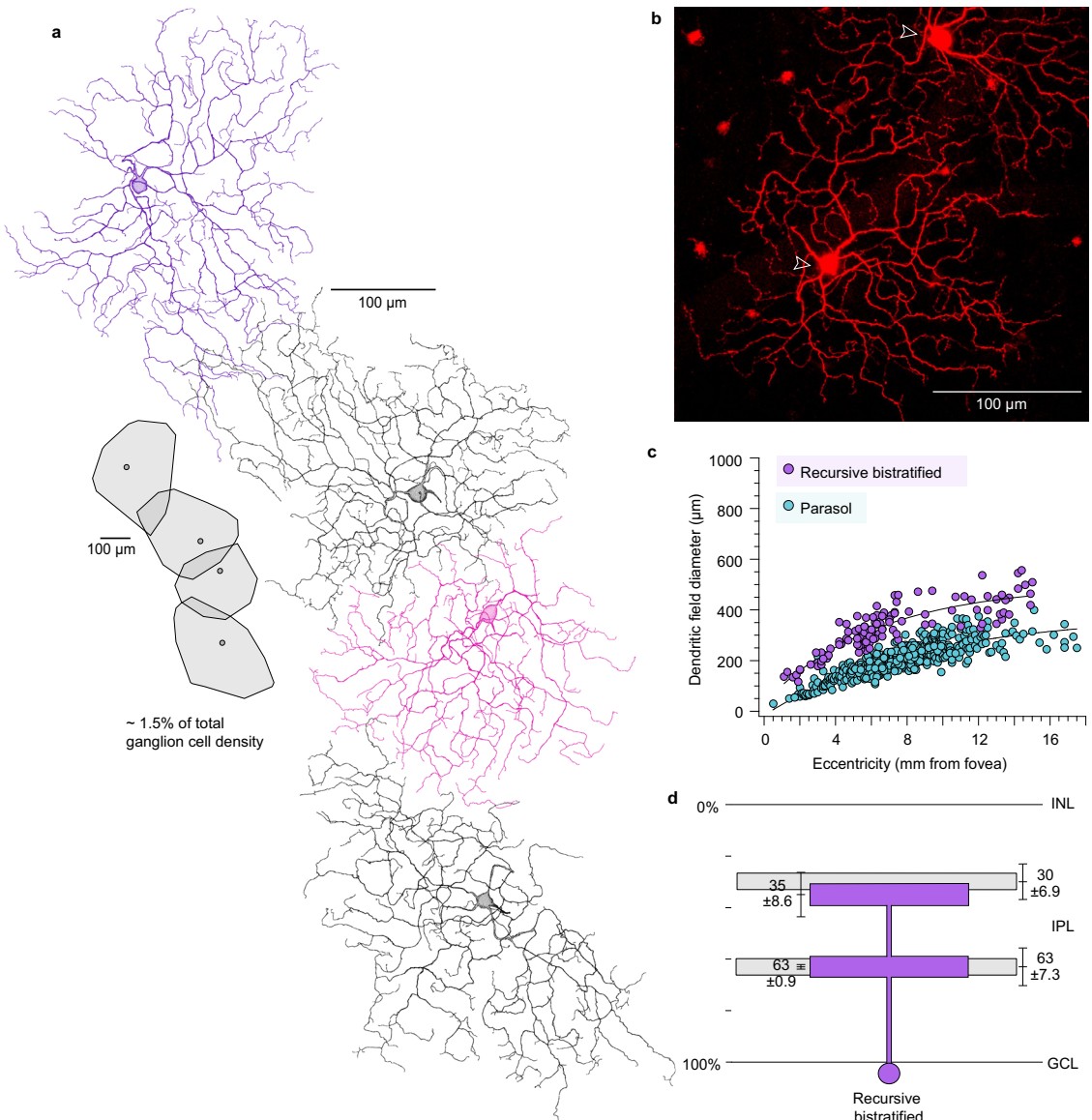

**Fig. 2 Dendritic morphology and field size, spatial density, and stratification depth of recursive bistratified ganglion cells in the macaque monkey retina.** Recursive bistratified cells comprise ~1.5% of total ganglion cells in the retinal periphery (dendritic field coverage = 1.3; see Supplementary Fig. 1 for details) and were retrogradely labeled and photostained in vitro from injections of rhodamine dextran made into the Lateral Geniculate Nucleus (LGN) and Superior Colliculus (SC). **a** Camera Lucida tracings of four photostained cells at ~7 mm retinal eccentricity from tracer injections in the superior colliculus; inset, outlines around dendritic perimeters for each of the four cells indicates regular spacing and little dendritic field overlap. **b** Photomicrograph of a recursive bistratified cell photostained in the in vitro retina after retrograde transport of rhodamine dextran permits precise targeting of this cell type for physiological study. **c** Dendritic field diameter of low-density recursive bistratified ganglion cells (purple circles) plotted as a function of retinal eccentricity ($n = 122$; mean ± s.d. = 327 ± 93; range = 117–557 µm); is large relative to a sample of LGN-projecting parasol ganglion cells, shown for comparison (blue circles). **d** Stratification depth of inner and outer recursive bistratified cell dendrites indicates costratification with the choline acetyltransferase immunolabeled strata ($n = 3$; see also Supplementary Fig. 1f). Data are shown as mean ± s.d.; $n$ number of cells.

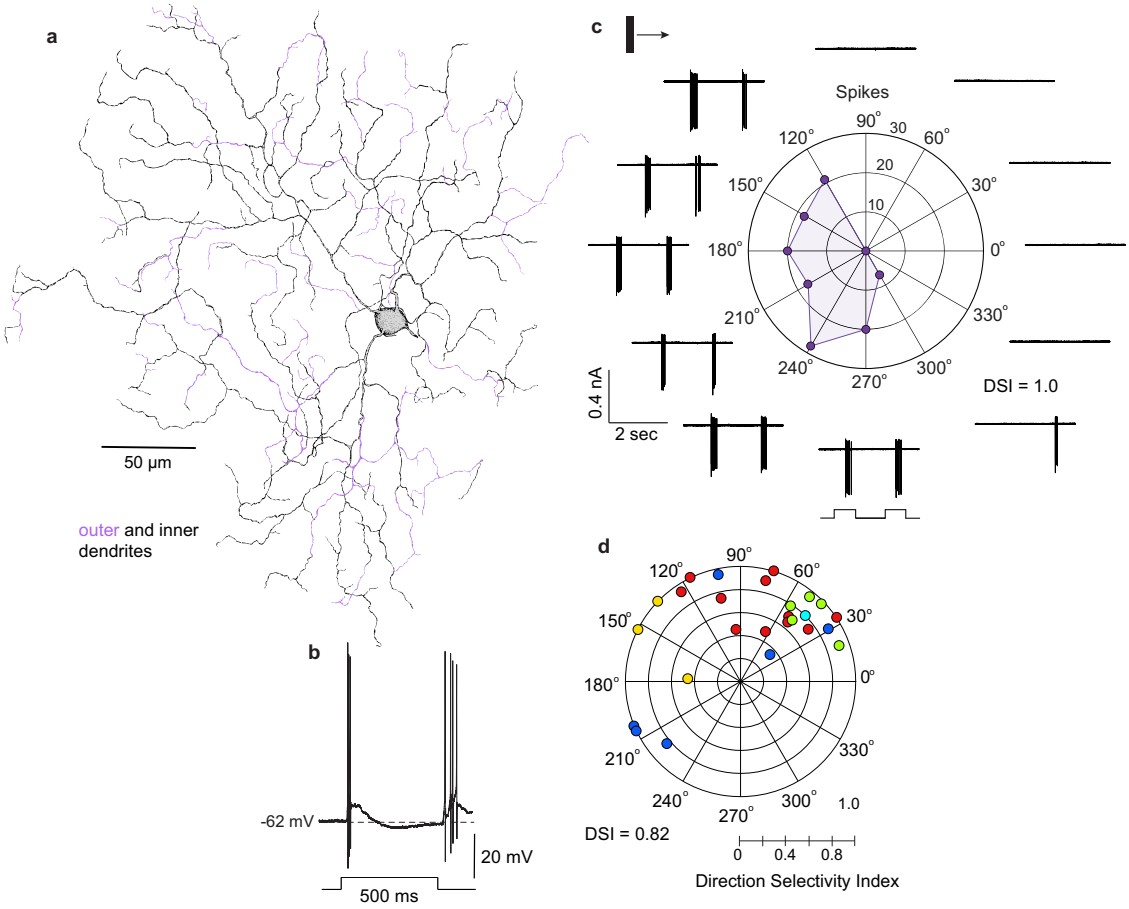

**Fig. 3 The recursive bistratified ganglion cell is an ON–OFF direction-selective type. a** Dendritic morphology of a recursive bistratified ganglion cell in macaque retina retrogradely labeled and photostained in vitro after rhodamine dextran tracer injections in the LGN. **b** ON–OFF response evoked by a 500 ms light step in a recursive bistratified cell targeted for whole-cell current-clamp recording identified by tracer coupling from an A1 amacrine cell (see Fig. 4 for details). **c** Polar plot of extracellularly recorded spike activity evoked by a bar of light (100 μm width, 600 μm height) moving (2000 μm/s) across the cell's receptive field. Spikes were summed from separate bursts of spikes evoked by 2 sweeps of the bar stimulus. **d** Plot showing Direction Selectivity Index (DSI) and preferred direction of spike activity evoked by a moving bar in 26 recursive bistratified ganglion cells extracellularly recorded from 5 different (color coded) retrogradely labeled retinas. The mean DSI ± s.d. of this data set was 0.82 ± 0.20 ($n = 26$). The stimulus parameters (bar dimensions, velocity, and contrast) were not standardized across all experiments, which may contribute to the cell-to-cell differences in DSI.

of the response that was being measured (spike rate vs total spike count) and the properties of the stimulus (e.g., velocity and contrast) that evoked it but did not explore these parameter spaces in detail here. In the mouse retina, variability in direction preference was linked to retinal topography and in turn to the optic-flow fields generated on the retina as the animal moves through the environment[8]. This question remains to be addressed directly in the primate, but such an optic-flow related spatial geometry predicts that as a population the ON–OFF DSGCs should show varied direction preferences aligned with the flow lines that radiate from the foveal center.

**The ON–OFF direction-selective ganglion cell is tracer coupled to a direction-selective, polyaxonal, ON–OFF spiking amacrine-cell type.** In some identified recursive bistratified cells, we made additional intracellular injections of the tracer Neurobiotin. Neurobiotin can pass through gap junctions and reveal cell-type-specific patterns of neuronal coupling that are distinctive for a particular retinal microcircuit[51]. In rabbit retina, such injections have shown that one of the four ON–OFF DSGCs is coupled to neighboring ganglion cells of the same type (homotypic coupling)[49]. For the macaque recursive bistratified cell, Neurobiotin coupling was not present in neighboring recursive

bistratified ganglion cells but was present in a distinctive cell type with a very large cell body that was often displaced into the IPL (Fig. 4a). The morphology, albeit incomplete, as revealed by the tracer coupling strongly suggested that these cells corresponded to the A1 amacrine cell, a previously identified spiking, ON–OFF cell type[52–54]. A1 cells showed a distinctive, polyaxonal morphology in which multiple, long-projecting axons arose from proximal dendrites and extended in various directions for millimeters within the IPL (Fig. 4f–h). To directly test the hypothesis that A1 cells are coupled to recursive bistratified ganglion cells, A1 cells in the in vitro retina were identified by their distinctive somatic morphology and transient burst of spikes at light ONset and OFFset[53] and were filled by intracellular injection with Po-Pro-1, a cationic fluorescent tracer that passes through gap junctions[55]. Po-Pro-1 revealed coupling patterns similar to that found for Neurobiotin but which could be observed by 2-photon (2 P) fluorescence imaging in the functioning retina in vitro[55]. Po-Pro-1 injections into A1 cells revealed both hetero- and homotypic coupling similar to that previously described[53], including coupling to a single ganglion cell type with a small cell body (Fig. 4b, c). The Po-Pro-1 labeled ganglion cells corresponded to the recursive bistratified ganglion cell type as hypothesized and showed the expected direction-selective light response (Fig. 4d, e).

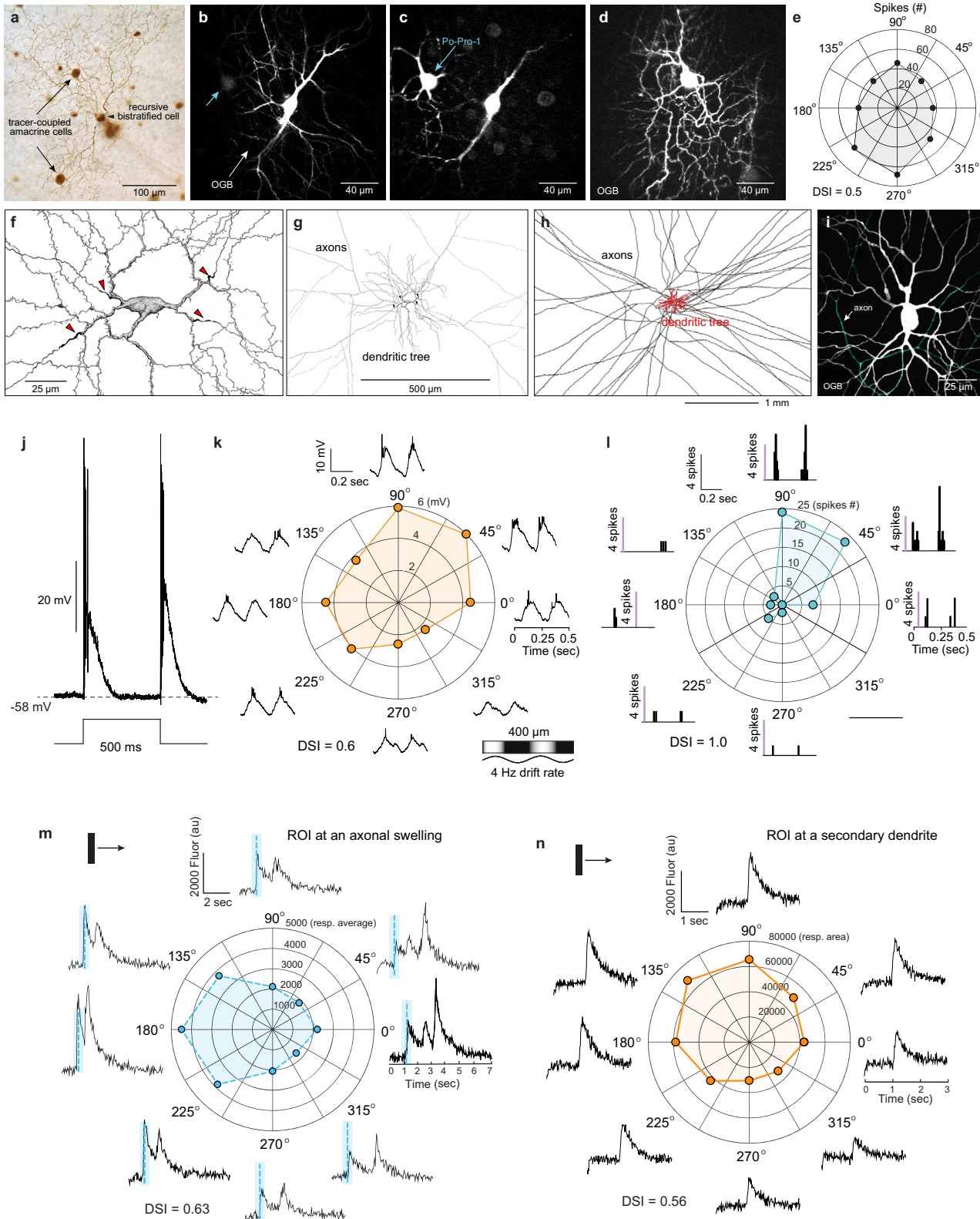

Given the link to the recursive bistratified cell and its similar transient, ON–OFF spiking response (Fig. 4j), we tested the A1 cell for direction selectivity. The membrane voltage recorded at the cell body showed directional tuning both for the modulation of the membrane potential (Fig. 4k) and the associated spike discharge (Fig. 4l) in response to moving bar stimuli. A1 cells were filled with the calcium indicator Oregon Green BAPTA-488

(OGB) during intracellular recording (Fig. 4i) and both the long-projecting axons and the thick, spiny dendrites showed stimulus-evoked direction-selective $Ca^{2+}$ signals (Fig. 4m, n, respectively).

**Connectomic reconstruction of A1 cells reveals polarization of synaptic inputs and outputs.** The A1 cell's distinctive axon-like

**Fig. 4 The recursive bistratified ON−OFF direction-selective ganglion cell is tracer coupled to the A1 amacrine cell, a polyaxonal, ON−OFF spiking cell type that also shows direction selectivity. a** A recursive bistratified ganglion cell (arrowhead; Neurobiotin fill) shows tracer coupling to the A1 amacrine-cell (arrows). **b** A1 amacrine cell (dye fill with OGB and Po-Pro-1) showing excitation of the OGB (488 nm). **c** Po-Pro-1 excitation (420 nm) a tracer-coupled ganglion cell is evident in the same field (blue-arrow). **d** OGB filling of a ganglion cell tracer coupled to another A1 amacrine corresponds to the recursive bistratified type. **e** Direction-selective response (intracellularly recorded spikes, DSI = 0.5) of cell shown in (**d**). **f** Drawing of an A1 amacrine cell shows the origin of four axon-like processes from proximal dendrites (red arrowheads). **g** The axon-like processes extend widely beyond the dendritic tree. **h** At lower magnification the full extent of the axon-like arbor (black) relative to the dendritic tree (red) is shown (Neurobiotin fills; **f, g** drawings modified from previously published anatomical descriptions of the A1 cell[47,48]). **i** OGB fill of an A1 cell in vitro illustrates how the axonal component (pseudo-colored blue−green) is distinguished from the main dendrites for functional calcium imaging. **j** The A1 cell responds transiently at light ONset and OFFset with large, ~60 mV spikes. Direction-selective changes in membrane voltage (peak to peak amplitude, mV) (**k**) and spike count (**l**) in the same cell in response to a sinusoidal drifting grating (4 Hz drift rate, 200 µm cycle period; 100% contrast). Peak to peak amplitude (mean DSI ± s.d. = 0.7 ± 0.21; $n = 5$) and total spike count (mean DSI ± s.d. = 0.47 ± 0.34; $n = 12$). Direction-selective calcium responses evoked by moving bar stimuli from A1 axon (**m**) and dendrite (**n**). For **m** (axonal swelling), the stimulus was a bright bar (500w × 1000 h; velocity 1000 µm/s; contrast 100%, (DSI = 0.63); 0.47 ± 0.10 (range = 0.28– 0.63), $n = 11$ axon ROIs). For **n** (dendrite), the stimulus was a faster moving, lower contrast bar (100w × 700 h, velocity 8000 µm/s contrast 60%, (DSI = 0.56); 0.41 ± 0.15, (range = 0.22–0.63, $n = 9$ dendrite ROIs).

processes (Fig. 4f–h) are obvious candidates for spike generation, long-range transmission across the retina, and synaptic output[54,56]. However, amacrine-cell processes can be both pre- and postsynaptic[57] and it is unknown how synaptic inputs and outputs are deployed on or from either the spiny dendritic tree or the long axon-like processes, making it difficult to hypothesize about the A1 cell's role(s) with regard to direction-selective circuitry. Thus, to begin to address this question we initiated connectomic reconstruction of an identified A1 cell (Fig. 5a). After an A1 cell was identified by intracellular recording and dye-filling, the retina was fixed for electron microscopy. Near-Infrared Branding[54,58] (NIRB) was then used to burn fiducial marks around the A1 cell body with the 2P laser. After embedding the cell was located by the NIRBed markings and scanning block-face electron microscopy[59] (SBFEM) was used to create a retinal volume that included dendritic and axon-like components of the identified A1 cell within the inner plexiform layer (Fig. 5b–h). We found that the thick, spiny dendrites were entirely postsynaptic, and indeed were embedded in a dense plexus of a near-continuous gantlet of inhibitory synaptic inputs from large distinctive varicosities (Fig. 5c, yellow balls, Fig. 5h) with bipolar ribbon synapses targeted to the prominent dendritic spines (Fig. 5d, red balls, Fig. 5h, zoomed inset). Partial reconstructions suggested that the dense inhibitory inputs to A1 dendrites include starburst amacrine cells. By contrast, the axons were entirely presynaptic, with the swellings along the axon's length making large vesicle-rich presumably inhibitory contacts (Fig. 5c, f–h; Supplementary Fig. 4). The majority output was to bipolar-cell axon terminals, with additional output to ganglion cell dendrites and a minor output to other amacrine processes (Supplementary Fig. 4). Thus, the spiking A1 amacrine appeared entirely polarized, with excitatory and inhibitory inputs to the spiny dendritic tree and inhibitory output arising from the distinctive synaptic boutons that studded the long-projecting, axonal processes.

**Morphological identification and direction selectivity of macaque starburst amacrine cells.** In non-primate mammals, the starburst amacrine-cell type is the synaptic origin of the direction-selective response observed in ON−OFF DSGCs. Thus, the direction selectivity found in the spiking A1 amacrine could also originate from the primate's starburst amacrine-cell type. It was therefore critical to identify starbursts in the macaque in vitro retina and to determine whether they show direction selectivity. We identified starbursts displaced to the ganglion cell layer (presumed ON-type), as others have in non-primate mammals[20], by their small, round cell bodies (~8 µm in diameter) and targeted them for whole-cell, patch-clamp recordings. All starbursts were confirmed by their highly stereotyped and characteristic dendritic

morphology after dye-filling (either OGB-488 or Alexa fluor 488 or 568) and correlated ON-type light response (Fig. 6a, b and Fig. 7a and Supplementary Fig. 5). Amacrine cells with some morphological features similar to starbursts were observed (Supplementary Fig. 5k, l) but distinguished from starburst cells by atypical physiological properties.

A key aspect of starburst cells' direction tuning is the preference for stimuli that move outward, or centrifugally, from its cell body[20,21,60]. Therefore, previous studies have used circular gratings centered at the cell body, that either expand (move outward) or contract (move inward) to measure direction selectivity in starburst cells[20,61,62]. In macaque, the somatic voltage response to such a 'bullseye grating' showed a large unequivocal preference for outward motion (Fig. 6c). To measure direction selectivity in the dendrites, as we did for the A1 cells, we again used 2P imaging of calcium transients after cell filling with OGB-488. Dendritic ROIs showed the same striking preference for centrifugal movement (Fig. 6d) as well as broad directional tuning to moving bars or drifting gratings (Fig. 6e–g)

**Macaque starburst amacrine cells show center-surround receptive-field structure independent of GABAergic inhibition.** In non-primate mammals, an inhibitory surround created by abundant starburst-to-starburst GABAergic synaptic connections has been proposed as a key mechanism for the centrifugal directional preference of starburst dendrites[60]. We thus wanted to further understand the receptive-field spatial structure of starbursts in the macaque to determine if an antagonistic surround was present. Using spots of light of increasing diameter or drifting gratings of increasing spatial frequency, we confirmed that inner starburst cells were ON center (Fig. 7a) and showed clear center-surround organization, well described by the Difference-of-Gaussians (DoG) receptive-field model[63,64] (Fig. 7b, c). Annular stimuli were then used to isolate the surround-response component which appeared as a large depolarization at light OFFset (Fig. 7d, shaded panel). Surprisingly, this isolated surround-induced OFF-depolarization persisted unattenuated in the presence of the GABAa receptor antagonist SR 95531 (GABAzine) (Fig. 7e, shaded panel). We hypothesized that the surround originates presynaptically in the center-surround organization of the excitatory bipolar cell's[65] input to the starburst. Further support for this conclusion is provided by somatic voltage-clamp recordings showing that synaptic currents evoked by annular stimuli that isolated the surround evoked inward (excitatory) currents (Supplementary Fig. 6).

If the starburst antagonistic surround arises presynaptically via excitatory bipolar input the surround would arise in the outer retinal circuitry, presumably by negative feedback from

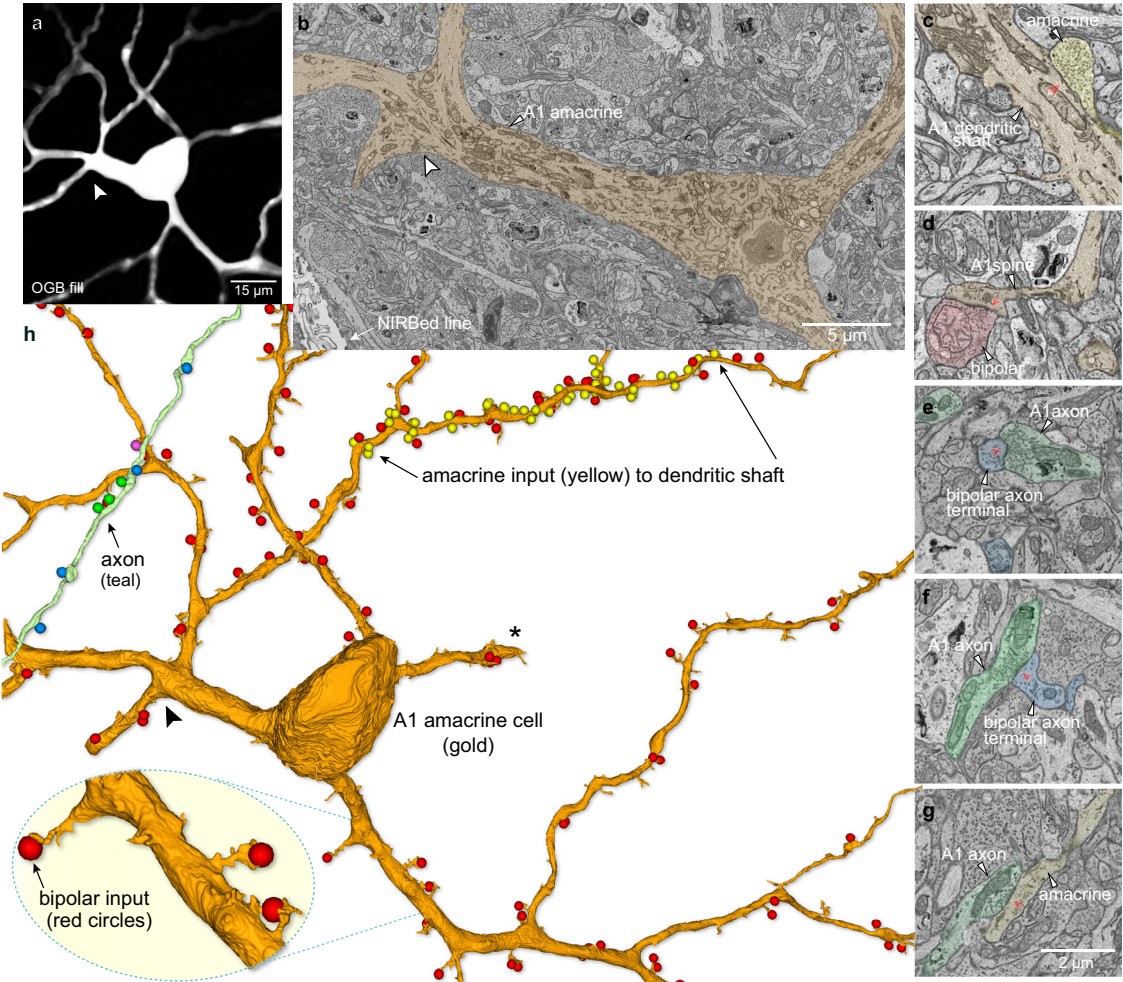

**Fig. 5 The spiking ON–OFF A1 cell shows complete polarization of synaptic inputs and outputs. a** Proximal dendritic morphology of a physiologically identified A1 amacrine cell, dye-filled (OGB + Alexa 468); 2P-confocal image; arrowhead points to the same dendritic location in **a**, **h**. **b** Ultrastructure of primary dendrites of A1 cell (tan fill) shown in a, with cell body protruding into the inner plexiform layer (IPL) confirms identification of this cell in EM tissue volume; partial view of NIRBed line at lower left (arrow). **c** A1 cell dendritic shaft (tan) receiving an inhibitory synapse (yellow, red arrowhead). **d** A1 cell dendritic spine (tan) receives a ribbon synapse (red arrowhead) from a bipolar-cell axon terminal (pink). **e**, **f** A1 cell axon-like process (teal) synapses on a bipolar-cell axon terminal (blue, red arrowheads) at 2 different locations; synapses from bipolar cells to A1 amacrine-cell axon-like processes were not observed. **g** A1 cell axon-like process synapses (teal, red arrowhead) on an amacrine cell process (yellow). The scale bar applies **c**–**f**. **h** Reconstruction of proximal dendrites and axon segment for A1 cell shown in **a**, illustrates that synaptic inputs from bipolar-cell ribbon contacts (red balls) were made primarily to the dendritic spines that arose from dendritic shafts (see inset lower left). By contrast, dense inhibitory synaptic inputs from other amacrine cells were made directly upon the dendritic shafts. Location of amacrine inputs to one segment of a secondary dendrite (between arrows) is indicated by the yellow ball structures. Reconstruction of a segment from the axon-like arbor (teal process at upper left) reveals synaptic output from varicosities primarily to bipolar-cell axon terminals (blue balls) but also to amacrine-cell processes (green balls) and to a ganglion cell dendrite (magenta ball). See also Supplementary Fig. 4.

horizontal cells to cones. We therefore tested the effect of HEPES buffer enrichment, which attenuates the surrounds of inner retinal neurons by acting to reduce negative feedback from horizontal cells to cones[66–68]. By contrast, with bath application of the GABAa receptor antagonist, the starburst surround was largely abolished by increasing the pH-buffer capacity of the Ames' medium bath solution with the addition of HEPES (pH 7.3; 20 mM) (Fig. 8a, b).

Given a presynaptic, excitatory basis for center-surround receptive-field organization in light-adapted macaque starburst cells, and the abolition of this surround by HEPES buffer enrichment we determined the effect of HEPES on starburst direction selectivity. HEPES dramatically reduced both somatic (Fig. 8c, d) and dendritic (Fig. 8e, f) preference for outward motion. For both the somatic voltage response (Welch's $F (1, 6.748) = 42.535$, $*p = 0.000382$) and

the dendritic calcium response (Welch's $F (2, 20.373) = 15.670$, $*p = 0.000076$), the effect of HEPES on responses to high-contrast, inward-moving gratings was primarily to increase their amplitude, with little effect on the response to outward motion. We conclude that, at least for radial motion stimuli, the bipolar cell's antagonistic surround is critical for starburst direction selectivity.

To develop insight into how the excitatory surround originating in bipolar cells was critical for radial motion sensitivity in starburst cells, we created a realistic neural model of the starburst receptive field based on the summation of bipolar cells' center-surround receptive fields to the dendritic tree. To do this, we first needed to estimate the receptive-field sizes of presynaptic bipolar cells as well as the identity and distribution of bipolar-cell synaptic input to the starburst tree. This was especially critical given recent evidence that starburst circuitry can vary

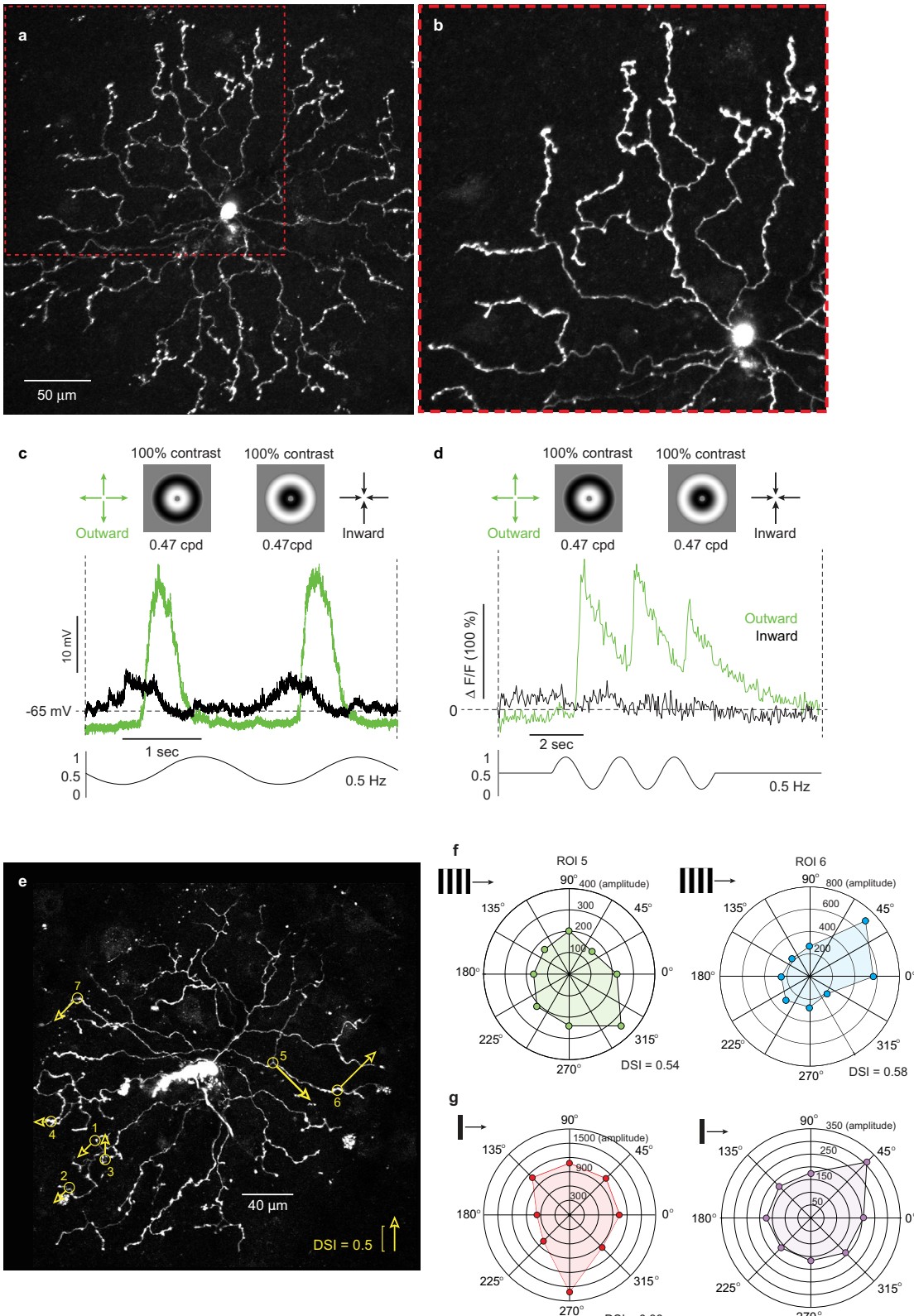

dramatically even across two well-studied non-primate mammalian species[69].

Stationary, contrast-reversing grating stimuli varied in spatial frequency can isolate nonlinear bipolar-cell "subunit" responses transmitted to Y-type ganglion cells by quantifying the second harmonic (F2) response[70,71]. We hypothesized that transient bipolar inputs to the starburst dendritic tree would likely show nonlinear spatial summation[48,72], and found that indeed such a stimulus revealed a second harmonic response in the starburst (Supplementary Fig. 7a) that extended to high-spatial frequencies, implying a small receptive-field center (30-50 μm Gaussian diameter, Supplementary Fig. 7b). We hypothesize that these subunits in the starburst receptive-field center correspond to rectifying bipolar-cell inputs[60,63]. Consistent with this hypothesis,

**Fig. 6 Morphological identification and direction selectivity of macaque starburst amacrine cells. a** Starburst amacrine cell targeted for patch-clamp recording in the in vitro retina and filled with the calcium indicator Oregon Green Bapta-488 (OGB). The thin primary and secondary dendrites and distinct varicose dendritic terminals create a radially symmetric, nearly circular dendritic tree, a feature common to starburst cells of other non-primate mammals (max intensity z-projection from 2P-confocal image stack). **b** Magnified view of boxed area (red dotted line) illustrates the characteristic extremely thin primary and secondary dendrites and the thicker, complex varicose terminal dendrites. **c** Somatic intracellular voltage recording of another starburst cell shows selectivity to radial outward motion using an expanding (green trace) or contracting (black trace) radial-grating stimulus; 2 stimulus cycles shown (100% contrast; 0.24 cycles/degree of visual angle; 0.5 Hz temporal frequency; n = 6; inward/outward = 0.26 ± 0.17). **d** Calcium response, 2 P optically imaged ROIs at the terminal dendrites of another starburst cell also show selectivity to radial outward motion; 3 stimulus cycles shown (n = 36 ROIs; inward/outward = 0.32 ± 0.18). Macaque starburst cell dendrites also showed direction selectivity to both moving bars and drifting square wave gratings. **e** Max intensity z-projection of OGB-488 filled starburst in vitro; locations of 7 dendritic ROIs are indicated by the yellow circles; arrows indicate preferred direction and arrow length indicates direction-selective strength (peak amplitude, mean of 3 stimulus cycles to drifting square-wave gratings (0.47 cycles/deg, 0.5 Hz) at 8 orientations; n = 7 ROIs; 0.29 ± 0.20; range = 0.15–0.58 DSI). **f** Polar plots of the optically imaged calcium response of 2 ROIs (5 and 6) from the cell shown in (**e**). **g** Examples of two additional polar plots for dendritic ROIs for two additional cells using drifting bars (bar w x h = 200 × 700 μm, 4000 μm/s) to measure directional tuning (n = 12 ROIs from 7 cells; mean ± s.d. = 0.37 ± 0.17, range = 0.12–0.76 DSI).

the distinctive F2 response to contrast-reversing gratings remained when the surround was attenuated by HEPES buffer enrichment (Supplementary Fig. 7c). Then, with a preliminary model of the starburst and an array of presynaptic bipolar cells, we found that a 30 μm bipolar cell receptive-field diameter could reproduce the real F2 response (Supplementary Fig. 7d). This calibration, along with connectomic data showing the number and type of bipolar cells converging onto the starburst, allowed us to further develop the model of the starburst receptive field

**Connectomic identification of cone bipolar input layout to a starburst dendritic tree**. Next, to determine the identity and spatial layout of bipolar input to the starburst dendritic tree we again used the NIRB method (Fig. 9a, b) and SBFEM to target a physiologically and anatomically identified starburst for microcircuit reconstruction. The central 150 μm of an OGB-filled starburst dendritic tree was completely reconstructed (Fig. 9a, g). Inhibitory synaptic outputs, and excitatory and inhibitory synaptic inputs to the starburst dendrites, were identified in high-resolution images and mapped (Fig. 9c–f). Ribbon synapses from bipolar-cell axon terminals were sparsely distributed over the dendritic tree (48 synapses in total) and occurred in the great majority of instances at small spines that extended from the very thin main dendrite (Fig. 9d, e, g inset). These spines usually contained small clusters of vesicles in proximity to the ribbon, but no clear synaptic specialization was apparent.

The complete morphology of the bipolar cells' axon terminals presynaptic to the starburst were reconstructed, and two anatomically distinct groups were identified. One group had a very small axon terminal with large, bulbous terminal swellings stratified in the IPL at the inner border of the starburst dendrites. The other group showed larger, more highly branched terminals with small terminal swellings stratified more superficially at the outer border of the starburst dendrites (Fig. 9h, i). The starburst dendrites thus traversed a space between these two bipolar axon terminal types, sending short spines either outward or inward to receive ribbon synaptic inputs from both groups. Complete reconstructions of the smaller, more deeply stratified terminals revealed extensive synaptic contact with identified midget (parvocellular LGN projecting) ganglion cells (Fig. 10a–e) and therefore could be unequivocally identified as the midget bipolar-cell type[73]. By contrast, the larger arbors made extensive synaptic output to an identified parasol (magnocelluar LGN projecting) ganglion cell (Fig. 10f–h). Only a single mosaic (single type) of these larger bipolar-cell axon terminals was evident but we refer to this type as DB4/5 because both previously described types (diffuse bipolar types 4 and 5) are now understood to costratify in the IPL, synapsing upon parasol ganglion cells, and showing largely indistinguishable axon-terminal morphology[73]. Thus, the

macaque starburst draws its excitatory input from the major parvocellular and magnocellular visual pathways that transmit sustained vs transient signals, respectively[74,75]. We determined the spatial layout of the inputs from these two bipolar-cell populations on the starburst dendritic tree (Fig. 9j), and found that the midget bipolar cells were distributed over the entire dendritic tree although tending to dominate proximally, whereas the DB4/5 axon terminals predominated more peripherally on the starburst dendritic tree (Welch's $F$ (1, 45.580) = 10.033, $*p = 0.002744$) (Fig. 9j; inset histogram).

Utilizing this information from our connectomic reconstruction, we built a realistic starburst model with 40 midget bipolars, providing a relatively sustained input to the starburst cell, located proximally out to a radial distance of 65 μm. Beyond this distance, midget and DB4/5 bipolar cells in the ratio 1/3 to 2/3, respectively, provided a more transient excitatory input (total of 70 bipolars) (Fig. 9k). Synaptic inhibition was omitted. The model replicated the experimental results, with the starburst cell strongly preferring outward radial motion (Fig. 9l). The model showed in addition that the proximal bipolar inputs to the starburst were, in particular, themselves outward motion-sensitive because their center response was unhindered by subtraction from a delayed surround (Supplementary Fig. 8). When the model was run without the bipolar-cell surround, none of the bipolars were motion-sensitive, and both proximal and distal bipolar cells showed negligible directional differences. This modeling suggested that the radial-grating stimulus generates directional differences in the starburst response based on the fundamental delay of the bipolar receptive-field surround. In models run with a bipolar surround with zero delay, the bipolar-cell responses showed no motion sensitivity, but a modest directional difference was generated in the starburst amacrine cell dendrites, suggesting that a given stimulus condition may engage varied redundant mechanisms for direction selectivity in the starburst amacrine cell. In this regard it is worth noting the critical interaction between surround delay and stimulus velocity. Thus, models with a smaller surround delay (e.g., 20 ms vs. 40 ms) produced a smaller but qualitatively similar outward directional preference in the starburst dendrites, that could be offset by models run with a higher velocity grating (e.g., 400 μm/s vs. 200 μm/s) which produced a larger outward directional preference.

## Discussion

Direction selectivity appears to start in the primate visual system much earlier than previously recognized. The retinal output has long been considered to be dominated by a few, relatively high-density visual pathways[76] that project via the parvocellular and magnocellular LGN to primary visual cortex[77]. It is now clear that these high-density pathways—linked to the requirements of high-spatial-resolution

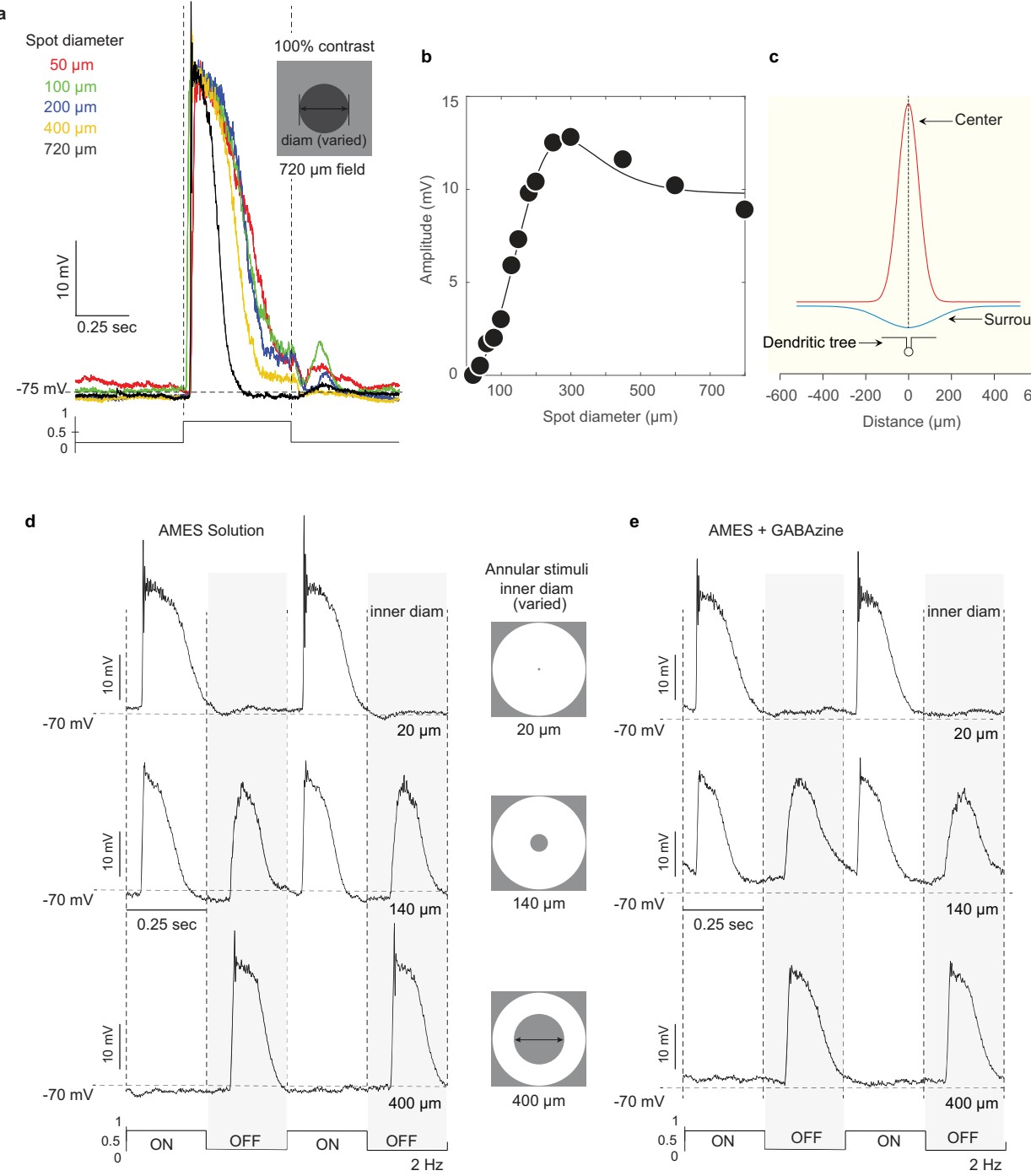

**Fig. 7 Macaque starburst amacrine cells show center-surround receptive-field structure that is insensitive to GABAa receptor antagonists.**
**a** Starburst whole-cell current-clamp recording (cell imaged in Supplementary Fig. 5e) shows a resting membrane potential ~75 mV on a high photopic background (~ $10^5$ photoisomerizations/cone/sec) and large amplitude depolarizing response (~40 mV) to 0.5 s, 100% contrast, square wave pulse stimulus. With increasing spot diameter (inset at top; 50–720 μm diameter; color coded) the response becomes more transient (black trace, 720 μm diam spot) and the sustained response component is lost. A characteristic feature of the starburst response to this stimulus is a large, spike-like transient that appears as the stimulus is enlarged to fill the receptive-field center. **b**, **c** Starburst spatial tuning (100% sinusoidal contrast modulation of spots of varied diameter at 2 Hz), fit with a center-surround Difference-of-Gaussians receptive-field model (line fit to data; $n = 27$ cells; center diam, $208 \pm 76$; surround diam, $360 \pm 156$; 2D profile of model fit is shown in **c**; small icon shows starburst soma and dendritic tree at same scale as model. **d** Surround response isolated by annular stimuli; increasing the inner diameter of the annulus from 20 to 400 μm (icons at center; top to bottom panels; 2 Hz square wave modulation) eliminates the ON-center depolarization (top panel) and isolates a pure OFF-surround depolarization (bottom panel). **e** Surround OFF depolarization is not eliminated by bath application of GABAa receptor antagonist (SR 95331(GABAzine); 10 μM; $n = 4$; $8 \pm 3$% reduction).

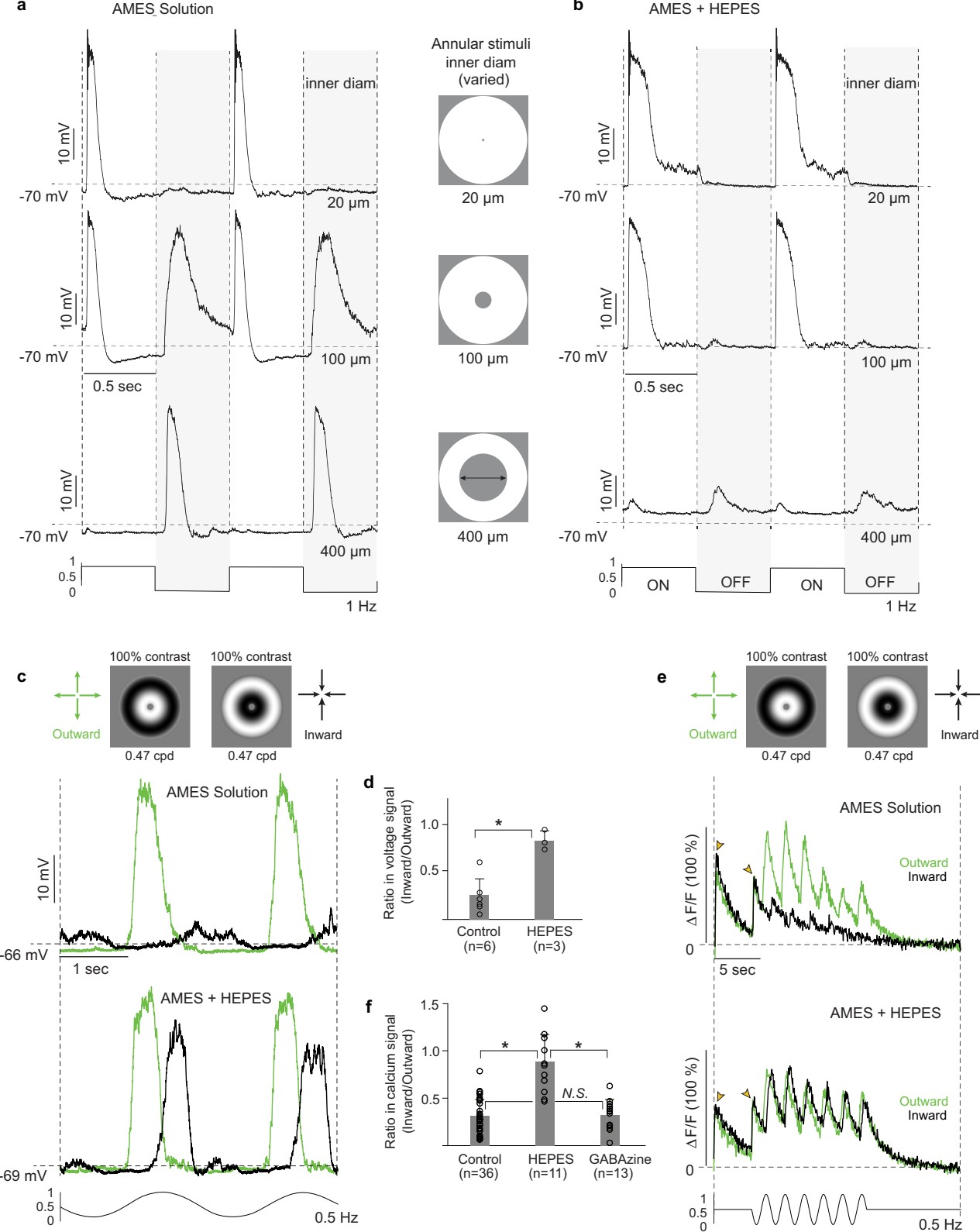

foveal vision—made it technically difficult to identify the great diversity of relatively low-density ganglion cell types, including discovery of chromatic[34,78], achromatic[47], and non-image-forming pathways[46]. The discovery of a direction-selective ganglion cell in the primate thus necessarily awaited a more complete accounting of these less accessible retinal cell types. Our catalog of 19 different RGC cell types accounts for ~97% of all RGC cells, leaving little margin for

additional types. This number is remarkably close to the number of RGC clusters identified in single-cell transcriptional profiling of both macaque[40] retina and human retina[45], but differs dramatically from the over 40 RGC types/clusters found employing similar methods to mouse retina[79].

The recursive bistratified ganglion cell was retrogradely labeled from tracer injections into both the LGN and superior colliculus

**Fig. 8 The starburst amacrine surround is attenuated by HEPES buffer. a** The same annulus stimulus protocol as shown in Fig. 7d (except that temporal modulation is 1 Hz) was used to isolate the surround mediated OFF response (top vs bottom traces; OFF response shaded). **b** HEPES buffer enrichment of Ames' medium greatly attenuates the surround OFF depolarization (top vs bottom traces, OFF response shaded) (20 mM HEPES; pH 7.3; n = 6; 83 ± 41 % reduction); **c** Top trace; somatic voltage response to radial motion (same protocol as Fig. 6c) in AMES solution shows strong preference for outward motion. Bottom trace, voltage response to radial motion in AMES with HEPES buffer enrichment (20 mM HEPES; pH 7.3). HEPES attenuates the preference for outward motion of radial-grating stimuli by increasing the response to inward motion. **d** Histogram plots inward/outward response ratio for control (n = 6) and HEPES (n = 3) application (unpaired t-test: t(8) = −3.3884, *p = 0.0095, Cohen's d = 3.76). Data are shown as mean ± s.d.; n, number of cells; N.S., no significant difference (p > 0.05). **e** Top trace, dendritic calcium response to radial motion stimulus; ROI at terminal dendrite; stimulus protocol as in **c** (see also Fig. 6d); 6 stimulus cycles are shown. Bottom trace, HEPES application again increases response to inward motion and eliminates outward motion preference. **f** Histogram plots inward/outward response ratio for control (n = 36), HEPES (n = 11) and GABAzine (n = 13) application. HEPES (Games-Howell post hoc test: *p = 0.000285, Hedges' g = 2.54) but not GABAzine (Games-Howell post hoc test: p = 0.984, Hedges' g = 0.05) eliminates the outward directional preference for radial-grating stimuli. Games-Howell post hoc tests were used to reveal statistical significance for three comparisons. Data are shown as mean ± s.d.; n, number of cells; N.S., no significant difference (p > 0.05).

and we make the parsimonious assumption that the single anatomically distinct population forming a single mosaic must project to both structures by a branching axon, as is the case for most other ganglion cell types. Only a small fraction (~10%) of LGN and collicular cells show direction selectivity[80,81], but this is the expected result for a very low-density retinal output that projects in parallel with a few high-density pathways. We observed a single ON–OFF DSGC population with unity coverage and even though we have accounted for greater than 95% of total ganglion cells (Supplementary Table 1) it remains possible that additional very low-density DSGCs remain to be discovered. It is possible that the known lower density of outer (presumed OFF) starburst cells in human and non-human primate retina could be explained by this relative reduction in ON–OFF DSGCs (1 type vs 4 types in mouse and rabbit). A population of recursive monostratified cells (Supplementary Fig. 1i, 2) are candidates for the ON-DS types present in non-primate mammals[7], but the visual physiology, and relationship to the cholinergic plexus for these ganglion cells has yet to be characterized.

The presence of direction selectivity in a spiking, polyaxonal, amacrine type linked to the recursive bistratified ganglion cell is distinctive but parallels an emerging picture in the non-primate mammal[6,69,82,83]. Direction selectivity in the ON–OFF A1 cell could arise via a gap junction with the ON–OFF DSGC (though we have not observed this feature in our connectomic reconstruction). If this were the case, the dominant axonal output synapse to bipolar-cell axon terminals could contribute to direction selectivity in the latter[83]. Alternatively, a directional signal could flow in the opposite direction, originating in the spiny dendritic tree. There is a striking precedent for such a circuit in mouse retina where a gap junction between polyaxonal amacrine cells and a single type of ON-DSGC underlies a direction-selective response that arises independent of GABAergic inhibitory circuitry[84]. Regardless of the origin of the direction-selective signal in the A1 cells, its spiking axonal projection suggests that a directionally-tuned inhibitory signal can be transmitted for millimeters across the retina (Fig. 4h), contrasting sharply with the local computation that occurs within semi-isolated electrotonic dendritic compartments of starburst amacrine cells[21]. Moreover, it remains unclear how the direction-selective output is represented in the multiple axonal projections that show distinct origins from separate dendritic sectors.

The neural mechanisms proposed for starburst dendritic direction selectivity include spatially asymmetric wiring of sustained and transient bipolar-cell inputs[85–87], starburst-to-starburst GABAergic inhibition[60], and the electrotonic properties[88] and ion-channel composition of the dendrites[61]. The asymmetric spatial layout of the bipolar-cell inputs from the classic sustained midget-parvocellular pathway and the transient parasol-magnocellular pathway to the macaque starburst bears some similarity to recent findings in the mouse[85,86]. However, the significance of such a synaptic motif as a mechanism for direction selectivity remains controversial[89] and, necessarily awaits further study in the primate. In the macaque starburst, a very strong surround was present, but surprisingly it was not mediated by the GABAa receptors on starburst dendrites, suggesting that inhibitory network interactions are not essential for light-adapted surround generation in primate retina[90]. Instead, we found that the surround arises presynaptically and is transmitted by the excitatory center-surround receptive field of the bipolar-cell[65] input to starbursts. Moreover, this excitatory surround is a critical element in the generation of starburst direction selectivity evoked by radial gratings, paralleling recent modeling of bipolar cells in mouse retina[91]. Our own initial modeling suggests that the delay of the bipolar surround relative to the center can be utilized to create radial motion preference in the starburst cell dendrites.

Individual dendritic sectors in mouse starburst amacrine cells give rise to direction selectivity in four ON–OFF DSGC types by highly selective asymmetric inhibitory connections[19]. The four DSGC subtypes differ in preferred direction and provide a directional signal that varies systematically with retinal location so as to align with optic-flow fields generated on the retina by the animal's motion through the environment[8]. In the primate, we have found only a single ON–OFF DSGC type. Outstanding questions thus include how these cells signal direction in relation to the optic flow organized around the foveal center in the forward-facing eyes of a primate[92–94], how this single, ON–OFF DSGC type is synaptically linked to the starburst dendritic plexus, and whether removing the bipolar-cell surround alters direction tuning in recursive bistratified ganglion cells. Lastly, our results raise the broader question of how the retinal direction-selective pathway interacts with direction and orientation tuned signals, generated in parallel, by similar circuit motifs[95,96] within diverse visual areas of primate neocortex[97,98].

## Methods

**Retinal in vitro preparation.** Eyes were removed from deeply anesthetized male and female macaque monkeys at the time of death (Macaca nemestrina, Macaca fascicularis, or Macaca mulatta) via the Tissue Distribution Program of the Washington National Primate Research Center and in accordance with protocols reviewed and approved by the University of Washington Institutional Animal Care and Use Committee. A total of 106 retinas acquired from 76 animals were used for this study. The retina was maintained in vitro by dissecting retina-choroid free of the vitreous and sclera in oxygenated Ames' Medium (A1420; Sigma Chemical Co., St. Louis, MO) under light-adapted conditions[78,99]. The retina-choroid was placed flat, vitreal surface up, in a glass-bottomed superfusion chamber coated with poly-lysine mounted, choroid side down, on the microscope stage. The retina was continuously superfused with Ames' medium (pH 7.3; oxygenated with 95% O2/5% CO2) and the temperature was thermostatically maintained within the chamber (TC-344B, Warner Instruments) at ~36 °C. The retina was observed under infrared illumination projected through the choroid from the microscope substage light source and visual stimuli were projected through the microscope optics through the objective lens.

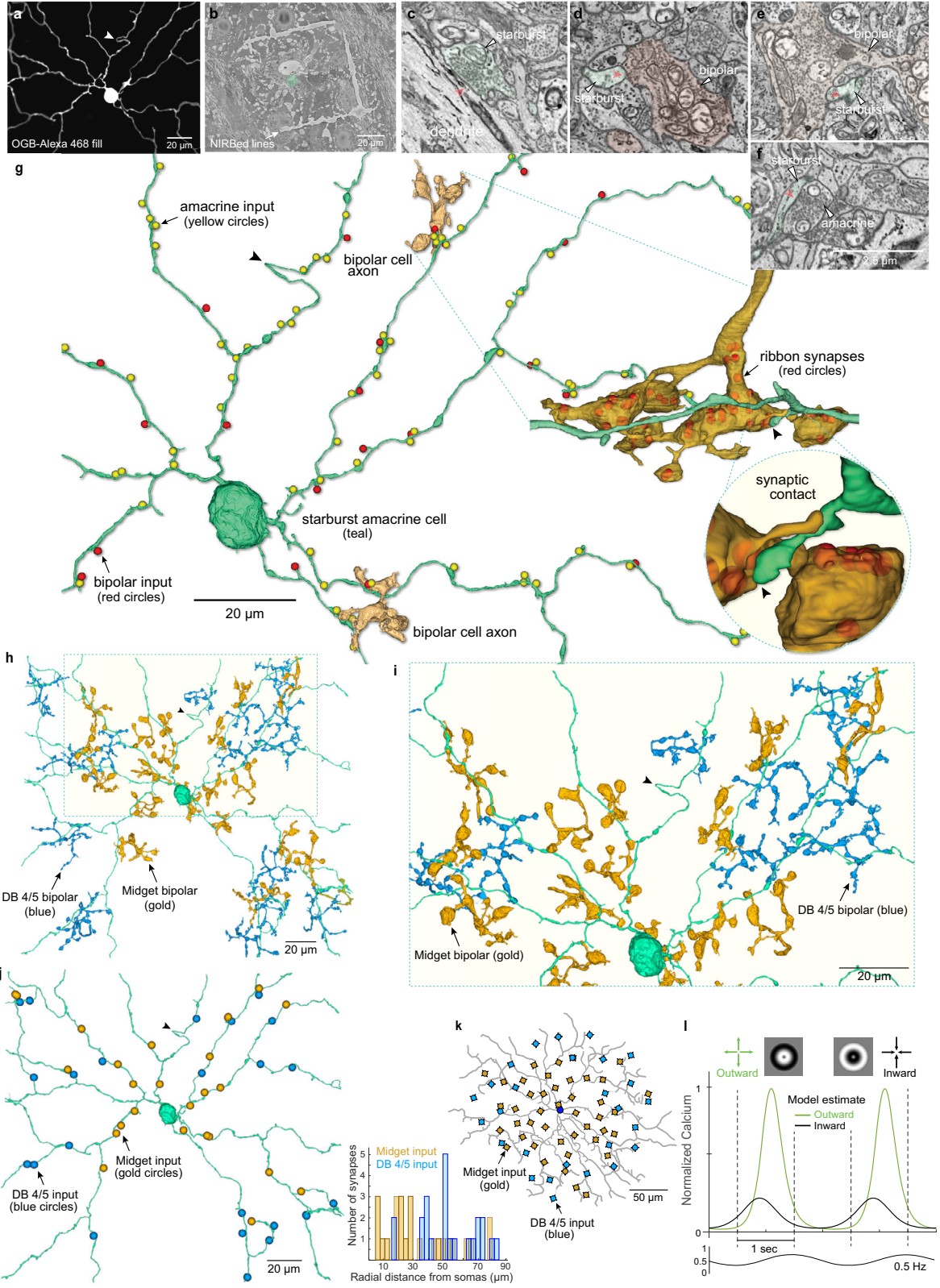

**Retrograde tracer labeling and in vitro photodynamic staining**. The tracer biotinylated rhodamine dextran was injected into physiologically identified central visual target areas (LGN, 14 animals; pretectum, 8 animals; superior colliculus, 7 animals) in adult macaque monkeys as described previously[47]. In brief, LGN injections were made after mapping LGN borders during recordings of large light-evoked multiunit potentials. Tracer injections were then made in both right and left LGN at multiple depths and anterior-posterior locations with the LGN boundaries. Similarly, SC injections were made in the visual responsive superficial layers at

multiple locations. While our histology showed injection sites confined to SC, we cannot exclude tracer spread to pretectal nuclei due to their proximity to the overlying SC. Our pretectal injections were made after recording visual activity in the pretectal olivary nucleus (PON); injections would have likely included the overlying nucleus of the optic tract/dorsal terminal nucleus (NOT/DTN) and would potentially encroach on fibers of passage projecting to the superior colliculus. The relatively large number of animals used for this study reflects progress through multiple experimental goals over an extended time period. For example, as

**Fig. 9 Connectomic reconstruction reveals spatial layout of distinct bipolar-cell types for an identified macaque starburst amacrine cell. a** Dendritic morphology of a physiologically identified starburst cell (OGB + Alexa 468 fill; arrowhead indicates hairpin loop in a dendrite (see also **g**). **b** NIRBing (see Methods) was used to burn fiducial marks in the optic fiber layer around the starburst cell (teal dot). **c** Starburst synaptic output from peripheral dendrite (teal fill; red arrowhead) to ganglion cell dendrite. **d, e** Bipolar cell (tan fill) ribbon synaptic input to short spines (teal fill, red arrowhead) arising from main dendrites. **f,** Inhibitory synaptic input to a proximal starburst dendrite (teal fill; red arrowhead; scale applies to **c–e**). **g** Inhibitory synaptic inputs (yellow balls) and bipolar-cell ribbon excitatory synaptic inputs (red balls) to a portion of the starburst dendritic tree (arrowhead indicates dendritic hairpin shown in **a**). The axon terminals of two presynaptic bipolar cells are shown (gold fills); one of these terminals is enlarged and rotated (dotted lines) with ribbon synapses indicated by red balls within the axon terminal. Inset shows how a starburst spine receives a single ribbon synapse from this axon terminal. **h** Two distinct bipolar-cell types synapse with the starburst tree; smaller axon terminals (gold) are midget bipolar cells and larger blue terminals are identified as diffuse bipolar type 4/5 (DB4/5) (see Fig. 10). **i** Zoomed view of all bipolar synaptic contacts on a portion (light yellow boxed area in **h**) of the dendritic tree. **j** Left, locations of all midgets (gold balls) and DB4/5 synapses (blue balls) on the starburst tree. Histogram plots distance of bipolar-cell axon terminal from the starburst cell body (midget bipolar, $n = 25$, mean ± s.d. distance from starburst soma = 34.5 ± 23.6 μm, DB4/5, $n = 23$, mean ± s.d. distance from starburst soma = 54.3 ± 19.6 μm; these mean spatial distributions differ significantly, unpaired t-test: $t(46) = -3.1433$, *$p = 0.0029$, Cohen's $d = 0.91$). **k** Distribution of bipolar inputs based on anatomical data (gold – midget bipolars; blue – DB4/5 bipolars) used in a starburst cell model (see "Methods") in which the cell received only excitatory input from simulated bipolar cells (5–15 pS/synapse; $n = 70$). **l** Distal dendritic calcium responses in the model showing robust directional difference to radial motion (see "Methods").

---

shown in Fig. 3, the recursive bistratified cells were targeted in retinas where photostained mosaics could be unequivocally identified. After a survival time of 4–7 days, animals were deeply anesthetized, the eyes removed, and the retinas prepared for in vitro physiological recording as described above. Labeled cells were photostained in vitro by brief exposure to epifluorescent illumination, which liberated the sequestered fluorescence tracer, revealing the complete dendritic morphology of labeled ganglion cells. Following an experiment, retinas were fixed and processed for horseradish peroxidase (HRP) histochemistry using the Vector avidin-biotin-HRP complex. Lightly labeled cells were enhanced in one of two ways. Either photochemically with 0.02% nitro blue tetrazolium (NBT; N-5514; Sigma, St Louis MO) to intensify the DAB reaction product, or by incubating the retinas in biotinylated anti-rhodamine (C# BA-0605, Vector Labs, Burlingame, CA) prior to HRP processing. Retinas were mounted flat using either an aqueous solution of glycerol and polyvinyl alcohol or Fluoromount-G (Cat. #17984-25; Electron Microscopy Sciences, Hatfield, PA), coverslipped, and stored refrigerated.

**Melanopsin immunohistochemistry.** Ten adult macaque retinas were processed for immunohistochemistry using a purified rabbit IgG against human/monkey melanopsin N-terminal (hNA) and biotinylated goat anti-rabbit (BA-1000; Vector Labs, Burlingame, CA); antibody design and characterization has been reported previously in detail[100]. Briefly, following enucleation, retinas were dissected free of the eyecup and fixed flat for 2 h in 4% paraformaldehyde. Primary antibody incubation was at 1:100 for 1–4 days and secondary antibody incubation at 1:100 for 1–2 days, both at 4 °C. After rinsing, retinas were processed for HRP histochemistry, intensified with 0.02% NBT when needed, mounted on slides and coverslipped as described above.

**Data analysis of HRP-stained cells.** Dendritic field area was determined from high magnification camera lucida tracings of individual cells. The perimeter of a convex polygon drawn around the dendritic tree was measured, and the area was calculated. Effective dendritic field diameter was taken as that of a circle having the same area. The tiling and dendritic tree overlap and thus coverage factor for each ganglion cell type was taken from clusters of photostained ganglion cells of the same type forming orderly mosaics. We then divided mean dendritic field area in mm² by coverage factor to determine cell density for a given type in cells/mm². This was then expressed as a percentage of the total ganglion cell number at this eccentricity (625 cells/mm²; 8 mm temporal retina)[89]. For large field low-density ganglion cells these estimates are little influenced by small variations in coverage. However, for the midget ganglion cells, which show an exceptionally high-spatial density, a small change in dendritic field overlap can produce a significant change in relative density. For this reason, our estimate that 97% of the total ganglion cells have been accounted for relies on the finding that midget ganglion cells show a coverage of ~1 (Supplementary Fig. 1b) which has also been shown in detail for the human retina[101].

**ChAT immunocytochemistry.** Ganglion cells were first targeted for intracellular staining using high resistance patch pipettes (>10 MΩ) with 2% Neurobiotin (Vector Laboratories) and 100 μM Alexa 594 (Invitrogen). After cell filling retinal pieces were fixed in 4% paraformaldehyde in 0.1 M phosphate buffer, pH 7.4, (PB) for 40 min. Immunocytochemical labeling was performed by using the indirect fluorescence method. Retinal pieces were incubated 4–6 days at 4 °C with goat anti-choline acetyltransferase (ChAT) antibodies (AB144P, Millipore) at a dilution of 1:200 with 5% normal donkey serum, 1% bovine serum albumin, and 0.5% Triton X-100 in PB. After washing in PB, secondary goat antibodies conjugated to Alexa 488 (Invitrogen) together with Alexa 568-conjugated streptavidin

(Invitrogen) to visualize Neurobiotin were applied for 3–4 h at room temperature at a dilution of 1:500.

**Light microscopy and IPL stratification measurements.** Image stacks of whole-mounted retina were taken with a FV1000 confocal microscope (Olympus) equipped with an argon and an He–Ne laser, using a UPlanSApo 60x/1.35 oil immersion objective at 1600 ×1600 pixels and a z-axis increment of 0.25 μm and processed with NIH FIJI software. The analysis of the stratification depths of ganglion cell dendrites and their position relative to the ChAT bands were usually performed at 4–6 different sites per ganglion cell's dendritic tree where the outer and inner borders of the inner plexiform layer (IPL) were determined and defined as 0% or 100% IPL depth, respectively. Subsequently, the stratification depth of the ganglion cell dendrites and the inner and outer ChAT bands were measured at the same location. This operation was performed for Parasol, midget, small bistratified, broad thorny and smooth monostratified cell types (see Supplementary Fig. 3). For the remaining types, where Chat band staining was not available, additional measurements (again at 4–6 sites where dendritic trees overlapped) were made relative to parasol and midget dendritic trees at the same location to determine IPL depth in relation to Chat bands. Vertical projections of image stacks from whole-mounted retina were created with Amira software (Thermo Fisher).

**Near-infrared branding (NIRB).** This method permits correlation of light and electron microscopy at the level of single identified neurons[58]. After an A1 amacrine or starburst amacrine had been targeted and identified physiologically and morphologically, the cells were intracellular filled with Alexa fluor 568. The tissue containing the cells was dissected and placed in EM fixative (2.5% glutaraldehyde/2% paraformaldehyde; in 0.1 M cacodylate buffer, 1 h). The tissue piece was then placed on a microscope slide, coverslipped in buffer and returned to the 2P-confocal microscope where the fluorescent amacrine-cell body was located and centered in the field of view with the appropriate filters under episcopic illumination. The cell body was then visualized with the 2P laser and scan line marks made with the focus on the retinal surface/optic fiber layer (linescan mode, 860 nm, power at the entrance to the microscope ~300–350 mW). Scan lines were 40–50 microns in length with scanner running at 40–50 Hz for ~10 s. With these parameters the scanned lines created tissue defects of ~ 2 μm in the XY and Z dimensions and were easily detectable after plastic embedding during preliminary semithin sectioning. Scan line defects in the retinal tissue was autofluorescent permitting images to be made in which the exact location of the scan lines relative to the fluorescent cell body are recorded; scan line defects located during subsequent sectioning thus permit a precise localization of the target amacrine cell.

**Serial block-face scanning electron microscopy (SBEM) and circuit reconstruction.** After NIRBing the tissue containing the target cell was then rinsed thoroughly in cacodylate buffer, pH 7.4 (0.1 M), and incubated in a 1.5% potassium ferrocyanide and 2% osmium tetroxide (OsO4) solution in 0.1 M cacodylate buffer for 1 h. After washing, the tissue was placed in a freshly made thiocarbohydrazide solution (0.1 g in 10 ml double-distilled H2O heated to 60 °C for 1 h) for 20 min. After another rinse, the tissue was incubated in 2% OsO4 for 30 min. The samples were rinsed again and stained en bloc in 1% uranyl acetate overnight at 40 °C, washed, and stained with Walton's lead aspartate for 30 min. After a final wash, the retinal pieces were dehydrated in a graded alcohol series and placed in propylene oxide for 10 min. The tissue was then plastic embedded (Durcupan, 44610, Sigma Aldrich). After NIRB lines were located during semithin sectioning the tissue block was trimmed, gold-coated by standard methods, and mounted in a Volumescope SEM (Thermo Fisher). The block face was scanned at 5 nm xy resolution (incident electron beam energy, 2.2 keV, pixel dwell time of 1.5 μs) after each of ~800, 50 nm

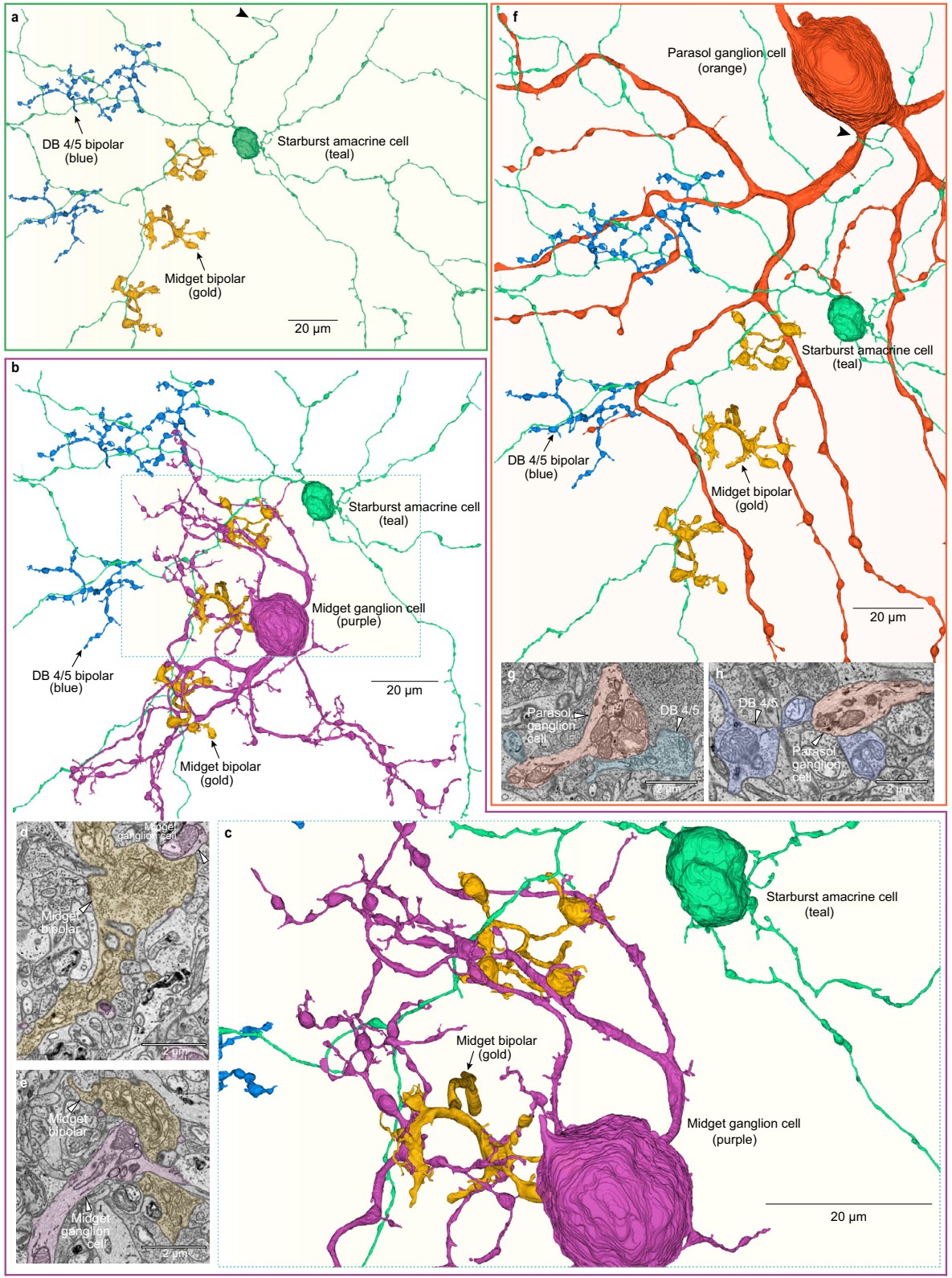

thick sections in an array of 20, 8000 × 8000 pixel (40 × 40 μm) tiles (10% tile overlap) that extended around the NIRBed area. The resulting set of ~16,000 TIFF images were contrast normalized, stitched into composite layers, then aligned into a volume using within layer and across layer procedures available with TrakEM2 software[102] (NIH FIJI plugin). All cell and circuit reconstructions were performed using TrakEM2 to first create arbor skeletons; nodes within these skeletons were placed on synaptic structures so that the locations and number of synapses for a given cell (e.g., number and locations of ribbon synapses in an

identified midget bipolar-cell presynaptic to a starburst amacrine cell) could be determined. Ground truth volume rendering of selected cells and cell profiles was performed manually using the painting tools available in TrakEM by multiple annotators and proofreaders.

**Electrophysiology and 2-photon imaging.** Retinal ganglion cells were observed using a 25X water immersion Olympus Objective (NA 1.05) under infrared

**Fig. 10 Connectomic reconstruction identifies midget and DB4/5 cone bipolar types as presynaptic to the starburst amacrine. a** Reconstructed starburst amacrine cell (teal; see Fig. 9) showing 3 cone bipolar cells with large axon terminals (blue) and three cone bipolars with small axon terminals (gold). The three large blue bipolar terminals are all presynaptic to the starburst. The two most proximal small gold bipolar terminals are also presynaptic to the starburst cell; the distal small bipolar overlaps but does not synapse with the starburst. Hairpin turn in starburst dendrite denoted by black arrowhead (see Fig. 9a, g also). **b** Complete reconstruction of a midget ganglion cell within the starburst dendritic field (midget cell identified by its uniquely small dendritic field size at this retinal eccentricity). The small gold bipolar-cell axon terminals are enmeshed by the ganglion cell dendrite and provide profuse synaptic input to this midget ganglion cell (25, 34, 27 ribbons synapses each from the three midget bipolar cells shown; this midget ganglion cell received a total of 405 ribbon synaptic inputs; the vast majority, ~90% were derived from midget bipolar cells) identifying them as midget bipolar cells. **c** Zoomed view of boxed area in **b** illustrates the close association of midget bipolar and midget ganglion cell for the two bipolars that also contact the starburst amacrine. **d, e** Two examples of ribbon synapses from midget bipolar (gold) to midget ganglion cell dendrites (purple). **f** partial reconstruction of an inner (ON center) parasol ganglion cell (orange; identified by uniquely large soma diameter) receives synaptic input from the 3 large (blue) bipolar terminals (~5 synapses/bipolar, 16 ribbon synapses total). Since inner parasol cells receive their major synaptic input from diffuse bipolar (DB) cell types 4 and 5 we identify this bipolar type as DB4/5 (these two types are not easily distinguishable (see main text). Hairpin loop in starburst amacrine (teal, black arrowhead) is created by wrapping around the primary dendrite of this parasol cell. **g, h** Two examples of ribbon synapses from the DB4/5 bipolars onto the dendrites of the parasol cell shown in (**f**).

illumination. Patch pipettes made from borosilicate glass were filled with either Ames' medium for extracellular "loose" patch recordings or with a K-based solution for whole-cell patch recordings. The pipette filling solution contained (in mM): K-gluconate (120), MgCl$_2$ (1) KCl (15), NaCl (8), HEPES (10), phospho-creatine, 2 K (7), ATP-Mg (4), and 0.3 GTP-Na (0.5), adjusted to pH 7.3 (Osmolarity, ~ 300 mOsm). In most instances 200 μM Oregon Green BAPAT-488, a fluorescent Ca indicator (150 or 200 μM) and/or Alexa fluor 488 or 568 (200 μM) was added to the pipette solution. We estimated a liquid junction potential of ~ 10 mV and therefore subtracted 10 mV from all membrane potentials intracellularly recorded in current clamp.

Data acquisition and the delivery of visual stimuli were coordinated by custom software running on an Apple Macintosh computer. Current and spike waveforms were Bessel filtered at 2 or 5 kHz and sampled at 10 kHz. To map the receptive field of the cell, the cell body was first placed at the precise center of the stimulus field. Flashing white squares (2 Hz temporal frequency, 10 or 25 μm wide) were systemically moved in the x and y directions to locate the most sensitive point of the receptive field and determine its approximate size. The location of the maximum spike response was defined as the receptive-field midpoint; and visual stimuli were positioned relative to this point.

Two-photon laser scanning fluorescence imaging was performed with the same 25x objective lens on a Sutter Instruments DF-Scope multiphoton imaging package for the Olympus BX51WI microscope that was modified to incorporate the output of a digital light projector that provided visual stimuli. Fluorescence was excited by 100-fs pulses at 80 MHz tuned to 935-nm center wavelength from a Ti:Sapphire laser (Spectra Physics Mai Tai HP). Laser intensity was regulated with a Pockels cell (Model 350–80 with model 302 driver, Conoptics) with the power measured at the retina of ~2.5 mW. ROIs for Ca$^{2+}$ activity scanning were 10 × 10 microns (34 × 34 pixels; 3.4 pixels/μm; pixel dwell time, 32 μs, 23.4 Hz frame rate). Fluorescence emission from the laser scanned fields was collected by the objective and delivered to a photomultiplier tube (PMT) (Hamamatsu GaAsP PMT, H10770PA-40) via dichroic mirrors and bandpass filters selective for fluorophore emission at 520 nm. The PMT output signal was digitized and processed using ScanImage, a software application for laser scanning microscopy (Vidrio Technologies). Laser scanning for morphological imaging of neuronal structure was typically done at either 512 × 512 or 1024 × 1024 pixels and image stacks were created at ~1.5 μm z resolution; maximum intensity flattened images were made with NIH FIJI software.

**Visual stimulus generation**. All visual stimuli used were created either by programming a computer graphics card (ATI Radeon HD 5770) in OpenGL or using a visual stimulator (VSG5, Cambridge Research Systems) equipped with a toolbox of stimulus configurations. The stimulus video signal drove a digital light projector (Vista X3, Christie Digital Systems) whose output beam was optically relayed into the microscope camera port where a dichroic mirror directed it down the optical axis and into the rear aperture of the microscope objective. The objective focused the stimulus onto the photoreceptor layer of the retina. The irradiance spectra for red, green, and blue light were measured with a spectroradiometer (PR705, Photo Research); peak wavelengths and integrated photon fluxes were 636, 550, and 465 nm and $5.7 \times 10^7$, $4.9 \times 10^7$, and $3.3 \times 10^7$ photons/s/μm$^2$, respectively. Light from these primaries were combined and with added calibrated filtering provided quantal catch rates for the long- (L) and middle- (M) wavelength sensitive cone photoreceptor types (comprising 90–95% of the cones) of ~6 × 10$^5$ photons/s/μm$^2$. These quantal catch rates are in the mid to high photopic range and we have previously shown that rods are in saturation at these light levels[31]. As previously quantified, given the uncertainties of the size of the cone aperture that strongly influence the efficiency of photoisomerization, we consider our estimates of quantal catch as very conservative. The intensity of the stimuli used in studies of human vision can also be expressed in units of retinal illuminance, or Trolands (Td) which scales by effective pupil size. We previously calculated that for a peripheral cone

with an inner segment aperture of 9 μm, 1 Td was equivalent to ~30 photo-isomerizations/s/cone[31].

**Direction selectivity measure**. For moving bars or drifting gratings a direction selectivity index (DSI) was calculated as, DSI = (P (Preferred response) − N (Null response))/P (Preferred response), where Preferred response is the maximum response and Null response is the response produced by the stimulus moving in the opposite direction. Responses were either total spike counts (extra- and intracellular recorded ganglion cells) or the peak response amplitude for stimulus-evoked dendritic/axonal calcium response in starburst and A1 amacrine cells. For spikes our criteria for direction selectivity was taken as at least a 50% difference in the null vs preferred direction response[103]. For example, a spike cell that responded to the preferred direction stimulus 50% more spikes than to the null direction stimulus would have DSI = 0.33. For the sake of comparison, a typical non-DS ganglion cell type like the midget cell had a mean DSI of 0.043 ± 0.03; $n = 4$ (ON cell type) corresponding to an insignificant 4% difference in the spike responses evoked by a stimulus moving in the preferred and null directions. For contracting (inward) vs expanding (outward) radially moving stimuli, we used the F1 Fourier value to quantify stimulus preference by taking the inward/outward response ratio. A ratio of 1 indicates equal response amplitudes to both inward and outward movement of the circular grating and a ratio 0.5 indicates that directional selectivity for inward is reduced 50% relative to outward motion.

**Statistical analysis and reproducibility**. All statistical tests were two sided and parametric. Statistical analyses were performed using SPSS version 28.0. In reporting and interpreting results, both the statistical significance ($p$ value) and substantive significance (effect size) were used. We used Welch's ANOVA, and follow-up Games-Howell post hoc test, paired/and unpaired t-tests on the data set when the sample sizes were unequal. We also report measures of effect size, such as *Hedges' g* and *Cohen's d* that reveals the magnitude of the difference between groups, since a $p$ value can only show that a significant effect exists but cannot reveal the size of the effect.

For the examples of light and electron micrographs of retinal cells and synapses (e.g., Figs., 2b, 4a–d and Fig. 5a–g) all images were reproducible and represent examples from multiple samples. For example, the tracer coupling from the A1 amacrine cell to the recursive bistratified ganglion cell in vitro was reproduced five times with images made. For all electron micrographs shown in Figs. 5, 9 and 10, all synaptic structures were identified (numbers provided in text) and the micrographs represent single examples from this large sample. Additional examples of A1 axon synaptic structures are provided in Supplementary Fig. 4. The starburst amacrine shown in Fig. 6a was selected from a large sample of images from all starbursts recorded and morphologically identified, with many more examples shown in supplementary Fig. 5. In the cases of our two retinal volumes created for connectomic reconstruction it should be noted that these comprise a single physiologically identified starburst amacrine cell and a single physiologically identified A1 amacrine cell and thus represent single examples of each of these cell types.

**Modeling**. The dendritic morphology used in the model was digitized from a confocal image stack of a dye-filled macaque starburst cell. The Neuron-C simulator[104] generated a compartmental model of the digitized morphology with a compartment length of 0.05 lambda for a total of ~300 compartments. The dendritic diameter at proximal, medial, and distal dendrite segments was determined from the charging current trace of an in vitro somatic voltage-clamped macaque starburst cell from a holding potential of −70 mV, to a +5 mV step. A least-squares fit of the model to the charging curve determined thickness factors for proximal (0.4 μm), medial (0.2 μm), and distal (0.8 μm) dendrite segments[89], consistent with direct measurements at the ultrastructural level. We modeled the excitatory input to the starburst cell as an array of bipolar cells ($n = 70$; density, 1900 cells/mm$^2$; average

spacing ~23 µm; regularity index 8). Most of the bipolar cells were given tonic release properties to simulate the midget bipolar-cell type. Beyond a radial distance of 65 µm, two-thirds of the bipolars were given a transient release property to simulate the transient release properties of presumed DB4/5 bipolars[105]. Although the DB4/5 bipolars had a more transient release than the midget bipolar cells, the major effect of the DB4/5 transient release in response to sine-wave radial stimuli at a low temporal frequency (0.5 Hz; see Fig. 9l) was to attenuate their input to the starburst cell compared to the more tonic midget bipolar-cell response.

Each bipolar cell was simulated as 2 compartments (one for the soma, and another for the axon terminal), and made a synapse onto the closest dendrite of the starburst amacrine cell if it was within a criterion distance (10 µm). Bipolar cells that did not make synaptic contacts were removed. Synaptic vesicle release was modulated by a readily-releasable pool that set the temporal properties of release (midget bipolar: pool size 50, max rate 200/s, DB4/5 pool size 8, max rate 50/s). The bipolar excitatory input synapses to the starburst were each given a maximum conductance appropriate to depolarize the starburst to −60 mV (Gmax, 1500; 5–15 pS per bipolar for typical stimuli). The postsynaptic effect of the lower release rate of the DB4/5 bipolars was increased with a greater maximum conductance (Gmax, 3000 pS per bipolar).

To simulate the starburst receptive field and responses to visual stimuli, we gave the bipolar center and surround Gaussian profiles (center diam 30 µm, surround diam 120–240 µm, surround/center weight 0.9–0.5) and ran simulated experiments using a variety of visual stimuli, including an expanding series of flashed spots, an expanding series of flashed annuli (outer diam fixed at 1000 µm), and linear and radial sine-wave gratings. Stimuli were presented to each modeled bipolar cell as a voltage pulse corresponding to the stimulus pattern. The mean intensity and contrast were given as voltages (mean ~ −50 mV, contrast 4–7 mV). The results from were then fit with a Difference-of-Gaussians (DoG) receptive-field model and compared with the equivalent DoG measurements of starburst amacrines from experiments on real cells. Typical Gaussian diameters from model fits for the simulated cells were ~160 µm for the center, and 300–400 µm, weight 0.4–0.7 for the surround.

To provide intuition about the mechanisms involved in starburst direction-selective responses, we included dendritic voltage-gated ion channels (K, Ca-T, NaV) in different combinations. Ion channels were simulated with Markov diagrams taken from the literature and included in the simulator. Potassium channels were modeled as a non-inactivating Kv3 channel[106,107]. Calcium channels were simulated with a slowly-inactivating T-type channel that approximated the fast activation and slow inactivation kinetics of N/P/Q-type channels[108,109]. Sodium channels were modeled as a NaV1.2 channel or a persistent NaV1.8 channel with slow depolarized activation kinetics[108,110,111].

To capture the appropriate center-surround dynamics, the bipolar surround Gaussian was simulated with a delay (typically 20–40 ms) from the center Gaussian with a strong integrated weight (0.8–0.9 of the center weight). While we recognize shorter surround delays in some previous models of primate ganglion cell receptive fields[e.g.112] modeled surround delay was derived from measurements of HEPES-sensitive hyperpolarization response latencies in macaque starbursts. The delay and the strong surround weight allowed the surround to subtract from and reduce the amplitude of the overall bipolar-cell response. Outward-moving radial gratings (spatial period 400 µm, velocity 200 µm/s, "contrast" 7 mV) evoked a ~10 mV response in proximal bipolar cells but inward-moving gratings evoked only a ~4 mV response, producing a directional difference of 6 mV (Supplementary Fig. 8). More distal bipolars had a smaller response to outward motion of the grating and a smaller directional difference (0.5 mV). A stationary counterphase grating (2 Hz) centered on the starburst soma at a series of different spatial frequencies evoked a frequency-doubled response comparable to the F2 plots from real data.

### Model biophysical properties

| Channel | Soma | Prox | Medial | Distal | |
|---|---|---|---|---|---|
| Kv3 | 1–3 | 0.5–2 | 0.2–0.5 | 0 | mS/cm$^2$ |
| Ca-T | 0.2 | 0.2–2 | 3–7 | 6–8 | mS/cm$^2$ |
| Ca pump Vmax | 2e−6 | 0.5-1e-6 | 1e-6 | 2e−6 | mA/cm$^2$ |
| NaV | 0.2–0.5 | 0.2–0.5 | 0.2–0.5 | 0.2–0.5 | mS/cm$^2$ |
| $R_m$ | 50e3 Ohm cm$^2$ | | | | |
| $R_i$ | 200 Ohm cm | | | | |

**Reporting summary**. Further information on research design is available in the Nature Research Reporting Summary linked to this article.

## Data availability
The data that support the findings of this study are available from the corresponding authors upon reasonable request.

## Code availability
The Neuron-C simulation language was used to generate the models described in this study. This code is available at: https://doi.org/10.5281/zenodo.6495531. Included in this distribution is the realistic SAC morphology, the "retsim" retinal circuit simulator for

model generation and the "rsbac_primate" script file that ran multiple model jobs in parallel. The scripts for the models are in the subfolder nc/models/sbac_primate. Neuron-C compiles and runs under Linux and Mac OSX. An Ubuntu Linux image that runs under virtualbox (http://www.virtualbox.org) is available at https://doi.org/10.5281/zenodo.6495719.

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

## Acknowledgements

This work was largely supported by grants from the National Eye Institute (NIH NEI) to D.M.D. (EY032045), R.G.S. (EY022070), P.D.G. (EY018369) and by National Institutes of Health (NIH) Grant RR-00166 to the Tissue Distribution Program of the Washington National Primate Research Center (WaNPRC), grant P51 OD010425 from the NIH Office of Research Infrastructure Program to the WaNPRC. and EY01730 to the Vision Research Core at the University of Washington. Additional support from MICINN Programa de Movilidad Salvador de Madariaga (PRX16/00188) to F.V. and NIH (NIBIB) R21EB028069 to J.B.T., and a Christina Enroth-Cugell and David Cugell Fellowship to J.W. We thank Sharm Knecht for EM tissue preparation and managing connectomics data acquisition and Ursula Bertram for connectomic reconstruction. We also thank Chris English for assistance with tissue acquisition and Christine Curcio and Andreas Pollreisz for advice and discussion.

## Author contributions

Conceptualization: D.M.D., P.B.D., Y.J.K., R.G.S., J.B.T.; Methods and technique development: D.M.D., P.B.D., O.S.P., F.V.; Programming and modeling: R.G.S., J.W., J.B.T., O.S.P.; Connectomics data analysis: Y.J.K., D.M.D., O.S.P.; Other anatomical analyses: J.D.C., H.R.J. and B.B.P., and C.P.; Physiological experiments: Y.J.K., D.M.D, O.S.P., and J.D.C.; Retrograde tracing experiments: D.M.D., P.D.G., J.D.C., and F.R.R.; Antibody generation: K.-W.Y.; Data analysis and figure generation: Y.J.K., P.B.D., O.S.P., and D.M.D; Writing and editing: D.M.D., Y.J.K., P.B.D., J.B.T., R.G.S., F.V., P.D.G., K.-W.Y.; Supervision: D.M.D., Y.J.K., P.B.D., R.G.S., and J.B.T.; Funding acquisition: D.M.D. R.G.S, J.B.T., P.D.G.

## Competing interests

The authors declare no competing interests.
