## [Peer Review File · Nature Communications]

Origins of direction selectivity in the primate retinaREVIEWER COMMENTS

Reviewer #1 (Remarks to the Author):

General comments

This is an impressive paper that deals with a question that has puzzled visual neuroscientists who work in NHPs for some time: given the widespread existence of directional movement selectivity (DS) in retinal ganglion cells (RGCs) throughout the vertebrate lineage, how is it the DS has been so difficult to find in monkey retina? The answer given here is comprehensive and detailed, and includes: the detailed description of a population of DS cells in macaque retina that are morphologically similar to DS cells in other animals; a valuable and comprehensive description of the responses of so-called “starburst” amacrine cells known to be critical for DS; and as well the discovery and description of an unexpected DS amacrine cell that is associated with the DS ganglion cell circuit. The work is of very high quality and settles a large part of the question posed by the paper's title. My comments are all minor and concern points of clarification or qualification.

Details

- The paper is chock-full of information and some of it is rather tersely presented, and I dislike having to consult the supplementary material for so much important information. In particular some large subset of the extended data figures 7 and 8, reporting the properties of starburst amacrine cells, seems important enough to be in the main body of the paper, and efforts should be made to include it in or near fig 4.

- A simple question, perhaps -- in fig 1d, why are there 2 bursts of spikes on each trace? The bursts are too far apart to be the leading and trailing edges of the bar.

- The paper discusses the population of recursive bistratified RGCs as a single type, but the degree of DS shown in Fig 1e reveals a great deal of diversity. Some cells have a DSI of 1, while others in the same population have values as low as 0.35 and 4 of the 26 have values <0.5 and would therefore not even be called DS by standard measures. Might there be two populations in this sample, perhaps by intrusion of some other population of RGC with recursive dendrites? Some comment here is needed.

- I may have missed it, but are the preferred directions given in fig 1e correct with respect to the visual world? If so, the strong prevalence of upward preferences is striking and deserves comment. If these

cells all prefer upward motion then it is difficult to argue that they form a comprehensive system for visual motion analysis.

- The starburst modeling adds useful information but the model has many degrees of freedom and some account of the sensitivity of the model to the choices of parameters would be useful.

- As far as I can tell, all the analysis of starbursts was done on the inner (ON) starbursts, and particularly that subset whose cell bodies are displaced into the RGC layer. But aren't the outer starbursts in monkey retina less numerous? For a true on-off DS, the two populations should work in the same way, should they not? Can the paper say anything about outer starbursts?

- It would be helpful to have a summary diagram of the retinal circuit outlining the hypothesized relationships among the DS RGCs, the starbursts, and the A1 amacrine. In particular I found it hard to keep track of the way the long range A1 amacrine would interact with cells having local DS. A schematic would help.

- The final sentence (ll 414-417) is gratuitous. It is silly to argue that all the evidence for encephalization of DS processing in the monkey brain is overcome by demonstrating that 1.5% of RGCs carry some kind of DS signal. That's not enough to make the retina "smart" compared to the CNS.

Reviewer #2 (Remarks to the Author):

When I was a graduate student I learned that lower prey mammals, such as mice and rabbits, have direction-selective retinal ganglion cells because they need to be able to make a lightning escape. Primates, on the other hand, were said to lack direction-selective ganglion cells (DSGC) because they compute the trajectory of moving objects in their visual cortex. It may take a bit longer, but predators have the luxury of time.

Nonetheless, it has always been suspected that primates actually do possess direction-selective ganglion cells, buried among the immense population of midget, parasol, and small bistratified ganglion cells that serve the major perceptual visual pathway.

Here for the first time, Dacey and colleagues show in convincing fashion that there exists a class of retinal ganglion cells, with distinctive "recursive" bistratified dendritic arbors, that have an on-off response to moving bars that is highly directional. These neurons tile the retina as a single class, making up about 1.5% of the retinal ganglion cell population.

The authors explore the circuitry that accounts for the direction-selectivity of these ganglion cells by reconstructing their inputs. They show that AI amacrine cells connect via gap junctions onto DSGCs, and are in turn direction-selective. They also explore the physiology of starburst amacrine cells, and show that they are directional, and make connections onto A1 amacrine cells. Hence the title of the paper, "Origins of direction selectivity in the primate retina".

The great strength of this work lies with the beautiful combination of physiology and anatomy. Dacey pioneered the method of retrogradely filling ganglion cells with rhodamine dextran by injecting the tracer into target nuclei. In vitro he then uses high intensity light to induce the tracer to fill the cell in a Golgi-like fashion, yielding exquisite anatomical detail. The receptive field properties of the labeled cell can then be explored in conjunction with the anatomy. In this paper Dacey and colleagues go a step further with the anatomy, doing serial reconstructions at the electronic microscopic level of laser marked cells to figure out the retinal circuit underlying direction selectivity. This work is a tour de force, although I must admit that I got a bit lost amid the complexity of the retinal circuit and I thought that this effort was only partially successful.

I wish the authors had explored the responses of the ON-OFF DSGCs to static bars presented at different orientations, given the relationship between orientation tuning, direction selectivity, and stimulus velocity. Are the ON-OFF DSGCs also orientation-tuned?

The biggest shortcoming of the paper is that the authors are vague about the location of tracer injections made to fill the ON-OFF DSGCs. Normally, one would expect to see images of injection sites in a paper like this one. I long suspected that direction-selective retinal ganglion cells in primates might project to the accessory optic system (AOS), a group of nuclei populated with strongly direction-selective neurons. Tracer injection confined to the AOS might label predominately ON-OFF DSGCs, while injection into the LGN would label many different cell types, a tiny minority of which would be ON-OFF DSGCs. We are told that tracer injections were made into the LGN, SC and pretectum. Magno or parvo laminae of the LGN? Did the SC injection site include perhaps the closely adjacent nucleus of the optic tract? What is meant by "pretectum"? Uncertainty about the preferred target of the ON-OFF DSGCs (or if there is a preferred target) means that an important part of this story is missing.

Another curious feature of the manuscript is that it seems to be a mixture of two articles: 1) an original, important, albeit incomplete account of direction selective neurons in the primate retina and 2) a review article updating the reader on Dacey and colleagues' recent work on classifying retinal cells in the primate. For example, Figs. 2 f,g are reproduced from earlier work by the author, and extended data figures 1, 2, 3, 4 are all tangential to the paper. Consequently, the paper is rather long and in some places feels like a hodge-podge.

Line 69: Does "These cells" refer to starburst amacrine cells or DS ganglion cells? Prose here is awkward, as starburst amacrine cells are formally introduced on line 72, after already being mentioned on line 68.

Line 77: Given that the authors observed back in 2003 primate RGCs with dendritic morphology similar to that of ON-OFF DSGCs in mice and rabbits, one is curious: what took them so long to establish the direction selectivity of these ganglion cells?

Line 121: It is stated that the 20% of ganglion cells not projecting to the LGN are various classes each present at low densities (not exceeding 1.5%). But aren't parasol cells most of the non-LGN projecting cells, and don't they comprise much more than 1.5%?

Line 124: What does "recursively branching" mean? Even after looking up the adverb, I'm not sure how it applies to these cells. What's the origin of the term -- did the authors coin it to describe these cells?

Line 127: It's interesting that ON-OFF DSGCs are filled after injections into "both the LGN AND SC." Do you mean "the LGN OR the SC"? If the former, doesn't it mean that the cells must send a branching axon to both LGN and SC. If the latter, doesn't that mean that separate populations of ON-OFF DSGCs project to LGN or SC, and they these populations tile the retina uniformly and completely a single time by somehow interdigitating perfectly?

Line 142: the mean DSI of the ON-OFF DSGCs is 0.82. Do you have data for the DSI of midget or parasol cells? I'm persuaded these cells are directional, but since the point of the paper is that the authors have discovered direction-selective cells in primates, data from a non-directional class of cells to provide a basis for comparison would be nice. Also, in addition to the strength of directionality, could the authors share data regarding its bandwidth?

Line 148. You don't know the "precise" location. Do you know the approximate location?

Lines 200-212: This description of the circuitry is complex, and really taxes the reader. You help by referring to yellow balls and red balls in describing inputs onto dendrites. Continue this assistance, by adding mention of blue balls, green balls, and magenta balls where appropriate. Do I have it right: A1 amacrine cells get input from bipolar cells and project back onto bipolar cells?

Line 220: We identified starbursts displaced to the ganglion cell layer, as others have in non-primate mammals . . ." It would be nice to add to this paper reference to Yamada ES (2003). Also reference her paper with Marshak DW (2005) – she identified a potential direction selective ganglion cell. Did she get it right?

Line 359-360: the authors suggest that the DSGCs they have identified may account for the directional selectivity of cells in the accessory optic system. Why didn't they inject the nuclei comprising the accessory optic system, to test this idea directly? Or maybe they did label the AOS ganglion cell inputs when they made "pretectum" injections, but didn't realize it? Along these lines, what is meant by "pretectum"? Do you mean nuclei governing the pupil response (e.g., olivary nucleus)?

Line 425: The reader understands that the 29 animals used for rhodamine dextran injection were among the 76 used for this study. If that is not true, please clarify. Some readers may wonder why it was necessary to inject so many animals with rhodamine dextran in order to retrogradely label ON-OFF DSGCs. Perhaps a bit of explanation might be useful, such as "a typical experiment allowed us to characterize the physiology of a single ON-OFF DSGCs; a total of 26 such neurons were studied in this report".

FIGURES:

In Fig. 1b, the ensemble of polygon sketches seems rotated clockwise with respect to the drawings of the actual cells. Could you orient them identically?* Where in the LGN was the injection made (parvo, magno?). What is the eccentricity of these cells?

*Moving on to Extended data figure 1, I now see that you often show a photomicrograph, drawing, and polygon sketch for cells, but each at a different orientation. Why not show each oriented consistently? It would be much easier for readers to make the transition between each representation of the data.

Fig. 1e, what does color signify? What does the single cell indicated by a star signify? The concentric rings in the plot represent the DSI, no? If so, why does it say "Spikes (#)" above the plot? Why the highly skewed distribution of favored directions for this sample of 26 cells? That's what you might expect if you kept hitting the same AOS subnucleus.

Line 896: leave out "near complete". There are 19 ganglion cells types. If mice have 40 types, this atlas is probably not "near complete".

Line 898: "a small fraction of the total ganglion cells that project to both LGN and superior colliculus". Do you mean that there is another cell type, besides recursive bistratified ganglion cells, that accounts for a much larger fraction of the cells that project to both targets? Do you mean sending a bifurcating axon to both targets?

Line 913: those are not tracings of the cells shown in a, but rather the left panel of b.

Can you detect any morphological difference between the parasol cells that project to SC versus LGN?

Is the cell in the left panel of Extended Data Figure 1c among the cells drawn in the middle panel?

Are the cells shown in Extended Data Figure 1f the same as those in Figure 1b, but now rotated and colored differently? If no other examples are available, and therefore you want to duplicate the illustration of these cells, please show them identically, don't mess around with them.

Can you explain the criteria used for the classification scheme presented in Extended Data Figure 1? Was it based purely on cell morphology, and dendrite stratification, relative to the ChAT bands in the IPL?

In Extended Data Figure 1, why does the distance between the lines representing the GCL and INL vary from panel to panel? Why does the width of the ChAT bands vary so much, and not always proportionately to the distance between the GCL and INL lines?

It seems like clutter to put 30 ± 6.9 and 63 ± 7.3 on every single panel.

Panel j, write "giant melanopsin cells" in place of "Giant MOPS".

There seem to be 17 types of RGCs in Extended Data Figure 1, not 19.

Line 1030: change "extened" to "extended".

Extended Data Figure 2: What is the point of including this figure? Are "Recursive monostratied" (do you mean monostratified?) cells direction selective? Why are these cells getting special attention in this manuscript?

Extended Data Figure 4: I'm also puzzled why this figure is included. It is cited on line 125 to support the fact that a rare ganglion cell type showed recursively branching dendrites that bistratified in the IPL, but the figure doesn't mention or show this rare ganglion cell type. Instead, it shows examples of parasol, midget, small bistratified, and broad thorny cells.

Extended Data Figure 5: it is amazing to capture the transition from dendrite to axon. Superb!

Extended Data Figure 7: caption says the resting potential is ~ 75 mV, but it looks like -65 mV.

Why are the schematics of the receptive fields shown at different sizes for a, b, and c?

In (a) the maximum spot size listed is 720 μm , but in the caption it says 800 μm .

Extended Data Figure 8: line 1132: legend seems garbled here.

Fig. 2: Are b and c showing the same field? What should the reader make of the difference in m (180 degrees) and n (130 degrees) in optimal direction for the slit? Are the dendrites and axons tuned to different directions? Would different dendrites in the same cell have the same preferred direction?

Fig 3: There seem to be many synapses visible in the EM (b) that are not shown in the reconstruction (h). Is that correct? Is (h) simply intended to provide examples of various types of synapses, or is it supposed to show all that are visible in the EM (b)? How are contacts with bipolar cells identified with surety? By relying on ribbon morphology?

Reviewer #3 (Remarks to the Author):

1. Kim and colleagues provide data of outstanding importance to understanding sensory processing in non-human primate (macaque monkey) retina. The data are also of high relevance to understanding human vision because macaques are evolutionary much more closely related to humans than the (nowadays) most common mammalian model system for visual processing i.e. mice. The brevity of this review does not reflect a lack of enthusiasm for the main result, which is that primate retina houses circuitry for direction selectivity (DS) largely aligned to that of better-studied mammals (mice, rabbits). In a trivial sense the broad result is confirmatory but at a deeper level it is very important because the presence of direction-selective circuitry in primate retina has been repeatedly called into question as part of the "two-or-maybe-three afferent channel" view that still dominates in textbooks. Overall the authors deserve high praise for this study, which in my opinion deserves rapid publication without too much nit-picking by DS cognoscenti.

2. The paper is for the most part very clearly written and the results are compellingly documented and illustrated. I did have trouble to connect the dots at some places. These are detailed below. My only major suggestion for improvement is to ask whether the authors really want to present the (stunningly documented) survey of non-DS cell types here, or rather publish it elsewhere. It will be buried alive in this paper, which is about DS, yet deserves to be accessed by a broad vision science readership. There is increasing interest, for example, from psychophysics modeling and retina prosthesis communities on primate retinal cell populations, but those potential readers will very unlikely stumble upon the valuable data presented here as extended figure 1. (The fact that the legend to extended figure 1 runs over five manuscript pages should I think give the authors pause for thought).

Minor:

3. I am not a member of the starburst fraternity therefore my comments to the starburst physiology do not come from a place of deep expertise. But even for a non-specialist there are some impediments to understanding. Starting from line 39 "generation of radial motion sensitivity ..." and popping up in many other places are strange circumlocutions and qualifications, where the authors seem to have doubts what relevance radial motion selectivity has for starburst responses to real-world stimuli. These are doubts which I share. The expanding ring stimulus does a brilliant diagnosis of centrifugal DS in starburst dendrites but such stimuli do not correspond to contours or movement in the external environment. Statements such as (l 399) "this excitatory surround is a critical element ... radial gratings" and elsewhere kept tripping me up: do the authors think that their conclusions about the bipolar-mediated surround apply to real-world stimuli or not? Some more clarity would be appreciated. Mention is made around line 336 of model responses to bars and gratings but data are not shown.

4. L76 / In the primate / In the primate retina /

5. L82-94 "distinctive ... unique ... Unexpectedly " [suggest remove or temper advertorials]

6. L91 "distinct antagonistic ..." I may be wrong, but at places it seems the authors want to emphasise differences between monkey and mouse / rabbit, but don't the similarities rather outweigh the differences? Mouse/rabbit starbursts also have a strong antagonistic surround to my understanding, and the evidence the authors provide that GABA is not involved in monkey is not terribly strong. They do show that a GABAa antagonist has negligible effect but that is only a single null result and they do not provide a positive control. Many readers would appreciate more balanced claims and discussion of limitations.

7. L102-104 I am sorry to be rude but this does seem a rather flimsy motivation for providing the encyclopedia of ganglion cells given in extended data figure 1. As noted above these data may deserve a better forum.

8.L107 "singular" [unclear]

9. L107 "photodynamic" [elsewhere the terms "photo-stained" (and on line 982 "photostained") are used, it perhaps it should be introduced here.

10. L205 / By contrast, the A1 axons /

11. L212 What is the conclusion/functional implication of these findings?

12. L233, L236 "surprisingly", "striking" [unclear; apart from the null gabazine result these data are quite compatible with mouse/monkey, aren't they?]

13. L328-345 This passage is quite impenetrable, it is not clear how the distinct temporal properties and/or spatial layout of bipolar inputs influenced the model results. The extended F11 does not show any parametric variation but rather two examples.

14. Line 1208 Surround delay 40 ms. This seems inconsistent with in vivo recordings e.g Smith et al., J Physiol 1992 found 5-10 ms surround delay.

15. L358 My recollection is that Hoffmann's group interpreted OKN as more retinal-driven but pursuit as cortical-driven, that would be worth following up. In any event this sentence is rather simplistic.

16. L1039 this gallery does not include a recursive bistratified cell.

17. L1204 -1206 this sentence is very unclear.

We thank the reviewers for taking valuable time to review our manuscript carefully and provide many helpful comments for improving the overall presentation of the results. We have addressed all of their specific comments point-by-point (noting changes to the MS) below. However, it is worth reviewing the major changes we made in response to the most general comments made by all three reviewers. Reviewer 1 thought that too much relevant data was placed in the supplementary figures making the paper hard to follow. Similarly, reviewer 2 had some trouble understanding the relevance or relationship between the original Main figures and the various supplementary figures. Finally, Reviewer 3 also felt that the supplementary figures related to our overall ganglion cell classification were “stunning” but their impact would be lost as supplementary figures. We understand their concerns and have addressed them by making major changes in the overall organization of the Figures. The revised MS now has 10 Main figures (in place of the original 5) and the supplementary figures now stand at 8 rather than the original eleven. We reorganized the presentation of the ganglion cell (GC) results into 3 new figures. Fig. 1 summarizes the overall classification and why this was important, Fig. 2 focuses specifically on the morphology and spatial density of the recursive bistratified GC type and Fig. 3 presents the direction selective data results for our sample of recursive bistratified cells. Figs. 4-5 concern the A1 amacrine cell and have not been changed from the originals. In response to Reviewer 1 we reorganized the starburst physiological data into 3 new figures (Figs. 6, 7, 8). We now walk thru this data beginning with the direction selectivity (Fig. 6), followed by receptive field structure and GABA antagonist data (Fig. 7) and finally the effects of HEPES buffering on receptive field structure and direction selectivity (Fig. 8) moving much of the previous supplementary data into the Main figure section, as suggested by reviewer #1. We also moved the supplementary Figure 10 (identification of bipolar type input to starbursts) to the Main Figures, now Main Figure 10. We should stress that no new data was added in rearranging these Figures. With this new layout, and the changes detailed below, we think that the paper has improved in clarity and we are grateful to the reviewers for pointing us in that direction. All specific changes related to this reorganization as well as those related to the minor comments made by each reviewer is given below.

Reviewer #1

General comments

This is an impressive paper that deals with a question that has puzzled visual neuroscientists who work in NHPs for some time: given the widespread existence of directional movement selectivity (DS) in retinal ganglion cells (RGCs) throughout the vertebrate lineage, how is it the DS has been so difficult to find in monkey retina? The answer given here is comprehensive and detailed, and includes: the detailed description of a population of DS cells in macaque retina that are morphologically similar to DS cells in other animals; a valuable and comprehensive description of the responses of so-called “starburst” amacrine cells known to be critical for DS; and as well the discovery and description of an unexpected DS amacrine cell that is associated with the DS ganglion cell circuit. The work is of very high quality and settles a large part of the question posed by the paper's title. My comments are all minor and concern points of clarification or qualification.

Details

- The paper is chock-full of information and some of it is rather tersely presented, and I dislike having to consult the supplementary material for so much important information. In particular some large subset of the extended data figures 7 and 8, reporting the properties of starburst amacrine cells, seems important enough to be in the main body of the paper, and efforts should be made to include it in or near fig 4.

As noted at the outset of this response we have taken these comments and those of the other two reviewers to heart and have moved as much relevant data as possible from the supplemental data to the main data figures (see related response to Reviewer 3 comments as well). With regard to the starburst amacrine cells we have created 3 figures (instead of a single figure 4) that walks through the critical starburst data by separating the DS data (new Fig 6) from the receptive field spatial data (new Fig. 7) and from the effects of HEPES buffer to attenuate both the surround and DS (new Fig. 8), moving much of the starburst data in the original supplementary figures to the main figures. In addition, we have moved all of the starburst connectomic data to the main figure set as well (new Fig. 10). To accomplish this, we have rearranged some figure elements for clarity but have not added any new data to our presentation. Manuscript text has been adjusted accordingly and should be self-evident.

- A simple question, perhaps -- in fig 1d, why are there 2 bursts of spikes on each trace? The bursts are too far apart to be the leading and trailing edges of the bar.

We thank the reviewer and agree that this data presentation was confusing. The two bursts represent two cycles of the stimulus and not the leading (ON) and trailing (OFF) responses to a single stimulus sweep since the bar was moving at a relatively high velocity (2000 $\mu\text{m}/\text{sec}$ or 10 deg/sec). We simply intended to show the response was relatively consistent across stimulus cycles where total spikes were summed to measure the directional response.

We now clarify this both in the Figure itself (new Fig. 3c) by adding a stimulus trace under spiking waveform and in the Figure caption as well.

- The paper discusses the population of recursive bistratified RGCs as a single type, but the degree of DS shown in Fig 1e reveals a great deal of diversity. Some cells have a DSI of 1, while others in the same population have values as low as 0.35 and 4 of the 26 have values <0.5 and would therefore not even be called DS by standard measures.

We agree with the reviewer that data plotted in figure 1e (now Fig. 3d) shows that degree of direction selectivity in our population of recursive bistratified cell is varied. We would, however, maintain that the observed cell-to-cell variability in DSI, which had a mean value of 0.82 ± 0.2 sd ($n=26$), is more likely due to the fact that the recordings were from five different retinas (now shown by the color coding in the figure) in which there are differences in the sensitivity of the in vitro retina that are related to the initial preparation of the tissue as well as total time in vitro. This of course does not rule out the possibility that there could be more than a single type of primate ON-OFF direction selective ganglion cell but one of the major findings in our overall results on the mosaic organization of the ganglion cell populations was that, 1) we were

able to account for nearly all ganglion cells and 2) across many anatomical samples (see new Fig. 2) we could only find a single morphologically homogeneous mosaic that tiled the retina uniformly with a coverage of 1, the agreed sine qua non of a single anatomical and functional population of cells. By contrast the recursive monostratified cells (the candidate ON-direction selective type(s)) showed a tiling pattern suggestive of at least 3 separate populations, similar to what has been observed in non-primate mammals.

We do not agree with the reviewer "that DSI < 0.5 would not even be called DS by standard measures". Note that a DSI of 0.35 (the smallest value in our data set) corresponds to the case in which the responses in preferred and null direction different by a factor of 1.54 meaning that the response in preferred direction has 54% more spikes than the response in the null direction. In agreement with Barlow and Levick (1965) our criteria for direction selectivity was based on a "just detectable difference" in the responses to a stimulus sweeping across the cell in opposite directions. And in Barlow and Levick the threshold for DS, i.e. the threshold of a "just detectable difference" was a $\sim 30\%$ difference in the preferred and null direction responses (see their Table 1 and text). Furthermore, with regard to questions about threshold DSI that serves as evidence of DS it is also relevant to point out that in a previous commonly cited study (Euler et al., 2002) the mean DSI for direction selective calcium signals in dendrites of starburst amacrine cells was 0.43 ± 0.04 SEM ($n=12$). The point being that the mean degree of direction selectivity in Euler et al was less than 0.5, which rebuts the contention that a $DSI < 0.5$ would not be considered DS by standard measures. In response to the reviewers comments we have revised the methods to state explicitly the criteria we have used for DS and also added a comparison with similar measurement from a typical "non-DS" ganglion cell type.

Might there be two populations in this sample, perhaps by intrusion of some other population of RGC with recursive dendrites? Some comment here is needed.

Our response to this question is covered above. But to expand further for clarity on this point, these cells were identified and targeted in vitro by their morphology and mosaic organization (see new Fig. 2a and b). The recursive bistratified cells formed a single mosaic (tiled the retina non-randomly with minimal overlap, interlocking like jigsaw puzzle pieces) (see Fig 2b for an example of the in vitro photostained mosaics that we targeted with our recording pipettes). We of course were expecting multiple types in which dendritic trees overlap extensively and fasciculate, as is the case for mouse and rabbit where four types overlap and their dendrites cofasciculate. Indeed this was a major reason we worked for an extended period to characterize ALL of the GC mosaics in order to determine if there were substantial populations that we could have missed (i.e., we wanted to address the question of whether we had accounted for the total GC density), which is why Figure 1 and supplemental data Figure 1 are important for our argument/conclusion that there is one ON-OFF type in primate while in mouse there are four.

- I may have missed it, but are the preferred directions given in fig 1e correct with respect to the visual world? If so, the strong prevalence of upward preferences is striking and deserves comment. If these cells all prefer upward motion then it is difficult to argue that they form a comprehensive system for visual motion analysis.

This comment now refers Fig. 3d in the revised MS. The apparent bias in our sample actually reflects the limited number of retrogradely labeled retinas (and limited number of retinal locations) in which we were able to clearly target the recursive bistratified cells (5 retinas; and 70% of the sample was derived from 3 of the 5 retinas) and the fact that we targeted them in patches (in most instances) at the same or similar retinal location (i.e., the same retinal quadrant) and they showed similar DS preferences. This is made clear now in the new Figure 2 where the cells from each retina are color coded.

A major unanswered question here – which this reviewer recognizes clearly – is that if there is only a single population of ON-OFF DS cells how are all directions represented. Recent work has shown that while multiple ON-OFF DSGCs are present in the mouse each type shows a complete range of DS preferences that depend on retinal location so as to conform to the direction of optic flow¹. From these results our expectation is that the DS preference of the single type of primate ON-OFF DSGC we have identified will depend on the location of the cell relative to the foveal center in alignment with the flow field lines that extend radially from the fovea. Thus, a single population of these cells may be all that is required for whatever analysis is performed at higher levels in the visual pathway. Our goal is to directly test this hypothesis in an ongoing study of a larger, more systematic sample of these cells where the cells location relative to the fovea is carefully recorded.

- The starburst modeling adds useful information but the model has many degrees of freedom and some account of the sensitivity of the model to the choices of parameters would be useful.

We thank the reviewer for commenting on the usefulness of the model. As we explain in our response to comments on the model made by review #3, the model was used to explore a limited set of experimental observations. We did explore the sensitivity of the model to the parameters that we varied. For example, when assessing the second harmonic response, we varied the size of the bipolar cell center to arrive at one that fit the experimental data (now noted on manuscript lines 303-305). For the case of the effect of the surround delay on directional signaling of the starburst for the radial-grating stimulus, we tested different values of the delay and different stimulus velocities (now noted on manuscript lines 364-367). As one might expect, the optimal rate of expansion/contraction for evoking a directional effect was related to the magnitude of the delay, but the effect was present for a range of physiologically likely delay times including those now referred to in the manuscript on line 365.

- As far as I can tell, all the analysis of starbursts was done on the inner (ON) starbursts, and particularly that subset whose cell bodies are displaced into the RGC layer. But aren't the outer starbursts in monkey retina less numerous? For a true on-off DS, the two populations should work in the same way, should they not? Can the paper say anything about outer starbursts?

Yes, all the experiments were focused on inner (ON) starbursts and this is the case for the great majority of the starburst physiology in other mammalian species because it is the ON starburst whose cell bodies are restricted to the ganglion cell layer and can be targeted under visual control for recording. In contrast, OFF starburst cell bodies are in the amacrine cell layer in the middle of the retina, a very difficult location to target in whole mount preparations like ours)

and can be targeted under visual control for recording. And, yes, there is some limited evidence that the OFF starburst population is lower in density than the ON (in Human retina, cell counts put the OFF density at about half that of the ON cells². In macaque it is less clear but the OFF population also seems slightly lower than the ON. However, the OFF ChAT bands are present and can be immunolabeled as shown in the MS (Suppl. Fig. 3). Our thinking about weaker Chat staining of the outer Chat band, given the present results, is that reduced density of the OFF relative to the ON starbursts may in fact be related to the reduced density of ON-OFF DSGCs in the primate (a single type) vs the non-primate mammalian models (4 overlapping types with identical stratification). By contrast there may be multiple populations of purely ON-DS cells (recursive monostratified type), though how/whether these cells interact with the inner ON starbursts remains to be investigated. We have now addressed this point briefly in the Discussion section (page 16, lines, 394-396).

It would be helpful to have a summary diagram of the retinal circuit outlining the hypothesized relationships among the DS RGCs, the starbursts, and the A1 amacrine. In particular I found it hard to keep track of the way the long range A1 amacrine would interact with cells having local DS. A schematic would help.

It is tempting to try to envision circuit diagrams but we felt that at this stage any proposed circuit diagram and functional architecture would be very premature/speculative and not very useful. The aim of this first paper is to introduce the major elements involved in DS circuitry in the primate. Future work will investigate thoroughly specific circuit motifs and related function. For example, it has already been proposed in the mouse retina that a second amacrine population with long, spiking processes, that is postsynaptic to starbursts and presynaptic to bipolar cells may be involved in either a type of “contextual” modulation of the DSGCs directional signal³. In the case of the primate retina future connectomic and physiological experiments directed at this question will be fruitful.

- The final sentence (ll 414-417) is gratuitous. It is silly to argue that all the evidence for encephalization of DS processing in the monkey brain is overcome by demonstrating that 1.5% of RGCs carry some kind of DS signal. That's not enough to make the retina "smart" compared to the CNS.

We thank the reviewer for pointing out the potential to confuse the reader in this last sentence. It was meant to simply reinforce the fact that the primate retina, a complex part of the CNS, constructs a specialized circuitry for DS which is therefore not encephalized. We recognize that the visual cortex has constructed DS circuitry, perhaps by similar mechanisms. We have replaced the concluding sentence with one that emphasizes this reviewer's more general perspective:

Lastly, our results raise the broader question of how the retinal direction selective pathway interacts with direction and orientation tuned signals, generated in parallel, by similar circuit motifs^{97,98} within diverse visual areas of primate neocortex^{99,100}.

As an aside, we would like to add that the low spatial densities of the various ganglion cell populations belies the importance of their functional roles in that this property mainly reflects a

lower spatial resolution/coarser grained map, which is not a critical feature for most ganglion cell types that do not underlie form vision. For example, the blue-ON color opponent cells are a small fraction of the total ganglion cells, with restricted projection to the LGN but presumably are critical for normal trichromatic color vision. Moreover, the likely presence of multiple populations of ON-DS cells – a focus of future work – suggests that the significance of these pathways for visual processing in primates may be somewhat underestimated.

Reviewer #2 (Remarks to the Author):

When I was a graduate student I learned that lower prey mammals, such as mice and rabbits, have direction-selective retinal ganglion cells because they need to be able to make a lightning escape. Primates, on the other hand, were said to lack direction-selective ganglion cells (DSGC) because they compute the trajectory of moving objects in their visual cortex. It may take a bit longer, but predators have the luxury of time.

Nonetheless, it has always been suspected that primates actually do possess direction-selective ganglion cells, buried among the immense population of midget, parasol, and small bistratified ganglion cells that serve the major perceptual visual pathway.

Here for the first time, Dacey and colleagues show in convincing fashion that there exists a class of retinal ganglion cells, with distinctive “recursive” bistratified dendritic arbors, that have an on-off response to moving bars that is highly directional. These neurons tile the retina as a single class, making up about 1.5% of the retinal ganglion cell population.

The authors explore the circuitry that accounts for the direction-selectivity of these ganglion cells by reconstructing their inputs. They show that A1 amacrine cells connect via gap junctions onto DSGCs, and are in turn direction-selective. They also explore the physiology of starburst amacrine cells, and show that they are directional, and make connections onto A1 amacrine cells. Hence the title of the paper, “Origins of direction selectivity in the primate retina”.

The great strength of this work lies with the beautiful combination of physiology and anatomy. Dacey pioneered the method of retrogradely filling ganglion cells with rhodamine dextran by injecting the tracer into target nuclei. In vitro he then uses high intensity light to induce the tracer to fill the cell in a Golgi-like fashion, yielding exquisite anatomical detail. The receptive field properties of the labeled cell can then be explored in conjunction with the anatomy. In this paper Dacey and colleagues go a step further with the anatomy, doing serial reconstructions at the electronic microscopic level of laser marked cells to figure out the retinal circuit underlying direction selectivity. This work is a tour de force, although I must admit that I got a bit lost amid the complexity of the retinal circuit and I thought that this effort was only partially successful.

We thank the reviewer for these positive comments and have addressed specific concerns and questions point by point below.

- I wish the authors had explored the responses of the ON-OFF DSGCs to static bars presented at different orientations, given the relationship between orientation tuning, direction selectivity, and stimulus velocity. Are the ON-OFF DSGCs also orientation-tuned?

While this is an interesting question that should be explored it is certainly beyond the scope of the current study the objective of which was to clearly and unequivocally identify major

elements of DS circuitry in the primate for future detailed study. A major question we are addressing now concerns the layout of ON-OFF DSGC direction preference in relation to the fovea to test our hypothesis that the preferred direction varies with retinal location and aligns with the optic flow field radiating from the foveal singularity.

- The biggest shortcoming of the paper is that the authors are vague about the location of tracer injections made to fill the ON-OFF DSGCs. Normally, one would expect to see images of injection sites in a paper like this one. I long suspected that direction-selective retinal ganglion cells in primates might project to the accessory optic system (AOS), a group of nuclei populated with strongly direction-selective neurons. Tracer injection confined to the AOS might label predominately ON-OFF DSGCs, while injection into the LGN would label many different cells types, a tiny minority of which would be ON-OFF DSGCs. Did the SC injection site include perhaps the closely adjacent nucleus of the optic tract? What is meant by “pretectum”? Uncertainty about the preferred target of the ON-OFF DSGCs (or if there is a preferred target) means that an important part of this story is missing. We are told that tracer injections were made into the LGN, SC and pretectum. Magno or parvo laminae of the LGN?

We have expanded our description of the tracer injections in the Methods section to address these comments (L468-478). All tracer injections were made after physiological recordings and mapping of light evoked mass potentials from either the LGN, superior colliculus or the Pretectal Olivary Nucleus (PON). In all of these studies our intent was to label as many GCs as possible – the LGN injections were made at multiple locations in both hemispheres including both magnocellular and parvocellular layers. We did map the boundaries of the LGN so as to restrict our injections to this structure. But our goal was to create as much labeling in the retina as possible from injections confined to the LGN and we did not attempt injections with laminar

specificity. One reason for this was the tremendous expense of these precious animals and we wanted to increase the chances of retinal labeling. We have previously published images of our LGN injection sites^{8,9} and our SC injection sites¹⁰. Our SC injections were made in the visually responsive, superficial layers; tracer seemed to be confined to the SC in all cases but we of course cannot rule out injection of fibers that also terminate in adjacent pretectal nuclei. Similarly, the pretectal injections were targeted to the PON based on physiological recordings. These injections would have likely encroached on the overlying NOT and potentially also labeled fibers of passage to the superior colliculus. But we really have no way of knowing the precise extent of the effective tracer uptake region even though we ran histology on these injection sites. The figure to the left helps to explain the close spatial relationship between the NOT/DTN with the PON.

Because we have shown injection sites in multiple previous papers focusing on different results related

to various ganglion cell types we felt that referencing these papers here would be satisfactory because the specific goal of this paper was NOT to define with high sub-nuclear laminar precision the central targets of the ON-OFF DSGC but to simply identify it as direction selective. Similarly, the goal of the ganglion cell type presentation was to calculate the relative densities of each type from their photostained mosaics. A deeper understanding of the precise targets of the ON-OFF DSGCs would, we believe, require many central recording experiments and very restricted tracer injections that are well beyond the scope of the present study.

Lastly, it is true that tracer injections into the LGN label many cell types and that the recursive bistratified cells comprise a small subset, and of course this was how we first identified this GC type in 2003. However, we did not have the expectation that ON-OFF DSGCs would form a major projection to any of the Accessory Optic System nuclei because all evidence from non-primates shows that another type(s) – the ON-DS cells - project to the AOS and that the ON-OFF DS cells project to the LGN and likely the superior colliculus.

In sum we are making no claims in this paper with certainty about the full set of targets of the ON-OFF DS cells – it is certainly possible that they could project to all three structures this reviewer mentions: the LGN, SC and the pretectal nuclei, NOT and/or DTN (AOS sub-nucleus). Indeed, we labeled these ON-OFF cells from injections into either the LGN and SC. Characterizing in a quantitative way the complete central projections of this cell type, its sublaminar specificity, whether it projects to any particular AOS subnucleus (LTN, DTN or MTN) is interesting but certainly well beyond the scope of this paper.

Another curious feature of the manuscript is that it seems to be a mixture of two articles: 1) an original, important, albeit incomplete account of direction selective neurons in the primate retina and 2) a review article updating the reader on Dacey and colleagues' recent work on classifying retinal cells in the primate. For example, Figs. 2 f,g are reproduced from earlier work by the author, and extended data figures 1, 2, 3, 4 are all tangential to the paper. Consequently, the paper is rather long and in some places feels like a hodge-podge.

Original Figure 2f-g, now Fig. 4f-g (we cite the origin of these modified images in the MS) are simply meant to introduce the reader to the known morphology of the A1 cell to set them up for the new results related to this cell type.

As we have reviewed at the start of our response we have tried to move much data from the supplemental file into the Main figures to show how it is important to the main conclusions of this study. We hope that this has made for an overall more coherent presentation.

-Line 69: Does “These cells” refer to starburst amacrine cells or DS ganglion cells? Prose here is awkward, as starburst amacrine cells are formally introduced on line 72, after already being mentioned on line 68.

“These cells” is now corrected to read: ‘ON-OFF DSGCs’ to make this reference clear to non-specialist.

Line 77: Given that the authors observed back in 2003 primate RGCs with dendritic morphology

similar to that of ON-OFF DSGCs in mice and rabbits, one is curious: what took them so long to establish the direction selectivity of these ganglion cells?

Initial studies dating to 2003 were purely anatomical with reference to the recursive bistratified cells. Experiments to combine morphology and physiology for these newly identified cell types was initiated ~2005 and due to restricted use and expense of primates extended through 2014. Physiological results from that time were held until we could develop new methods to target the ON-OFF DSGCs and to characterize the starburst amacrine to strengthen the overall significance of these results. It might also be noted (and this is now shown in the new Figure 2) that our sample of 26 ON-OFF DS recursive bistratified cells were derived from 5 animals with the majority deriving from 3 animals (see response to reviewer 1). These were extremely difficult experiments in which we attempted to find anatomically clear ‘patches’ or mosaics of a single type that were identifiable in vitro by photostaining. The use of the Po-Pro1 tracer-coupling described in our current results now permits more reliable and consistent targeting of this DS ganglion cell type without the need for the much more difficult and expensive central injection/photostaining approach.

- Line 121: It is stated that the 20% of ganglion cells not projecting to the LGN are various classes each present at low densities (not exceeding 1.5%). But aren't parasol cells most of the non-LGN projecting cells, and don't they comprise much more than 1.5%?

We apologize for the confusion here and have revised this sentence to clarify. We simply meant that the three major groups of high-density GCs (midget, parasol, and small bistratified) comprise 80% of the total ganglion cells and project to the LGN. The remaining low-density types have been retrogradely labeled from multiple locations, including the LGN and superior colliculus. Indeed, most types described in this study have been retrogradely labeled from tracer injections into the LGN.

Line 124: What does “recursively branching” mean? Even after looking up the adverb, I’m not sure how it applies to these cells. What’s the origin of the term -- did the authors coin it to describe these cells?

The term ‘recurve’ has been used previously¹¹ with regard to rabbit DSGCs to describe dendrites that curve back towards the cell body (rather than extend radially as in a typical alpha type ganglion cell). And it is well established that this recursive anatomical feature is dictated by fasciculation with the similarly looping branches of starburst amacrine dendrites. We did in fact coin the term “recursive” when we originally observed and named these cells in the primate⁸. We have now stated more directly (L 69-71) how recursive as an identifier or name for this cell type is used descriptively to denote the characteristic dendritic tree structure of these cells that has been observed previously in mouse and rabbit.

Line 127: It’s interesting that ON-OFF DSGCs are filled after injections into “both the LGN AND SC.” Do you mean “the LGN OR the SC”? If the former, doesn’t it mean that the cells must send a branching axon to both LGN and SC. If the latter, doesn’t that mean that separate populations of ON-OFF DSGCs project to LGN or SC, and they these populations tile the retina uniformly and completely a single time by somehow interdigitating perfectly?

We did not perform double labeling experiments with different tracers injected into LGN and SC. We found the recursive bistratified cells labeled from injections made into either the LGN or the SC in separate experiments (like the parasol-Y cells and many other types). Because there is a single mosaic of these cells (like the parasol-Y cells) our parsimonious assumption is that they project to both targets by a branching axon. This would be the common pattern for the great majority of ganglion cell types, with the exception of the midget ganglion cells that appear to only project to the LGN.

We have clarified this point in the text to now read: These cells were retrogradely labeled from tracer injections made into either the LGN or the SC (L129-130).

Line 142: the mean DSI of the ON-OFF DSGCs is 0.82. Do you have data for the DSI of midget or parasol cells? I'm persuaded these cells are directional, but since the point of the paper is that the authors have discovered direction-selective cells in primates, data from a non-directional class of cells to provide a basis for comparison would be nice.

We did not find direction selectivity in the other proposed candidate for DS in the primate – the broad thorny cell – but this result has already been published by others¹². We have run DS stimuli on parasol and midget cells and a number of other types but did not find DS. For example, the mean DSI for 4 ON midget cells we recently recorded was essentially zero (see an example of one typical midget ganglion cell below, DSI = 0.02). As mentioned in the MS we have not recorded from the recursive monostратified cells which we predict will correspond to the ON-DS type. Given 50 years of recording from primate ganglion cells with no reports of DS in the major LGN projecting types we did not think it necessary to add this to the present study. However, in the Methods section, in describing our criteria for DS (in response to a Reviewer 1 comment) we now add a summary of this midget cell data for comparison (L 655-661).

- Also, in addition to the strength of directionality, could the authors share data regarding its bandwidth?

We agree with the reviewer that the discussion of the properties of directional tuning in DS cells could include a parameter that expresses the sharpness of tuning in a way that is analogous to bandwidth (or more specifically the Quality Factor Q) in describing the sharpness of frequency tuning in an electrical filter. To do this we turned to directional statistics and fitted the polar plots of direction-dependent responses with the Von Mises distribution. This is the circular analog of a normal distribution having two parameters, μ , the preferred orientation, and κ (K), a measure of tuning dispersion, where $1/\kappa$ = the variance of the distribution. As with the Q of an electrical filter, tuning sharpness increases with increasing κ . This proved to be an appropriate way to express sharpness of tuning of individual DS ganglion cells, A1 amacrine cells and starburst amacrine cells, but it was not a useful means of comparing the sharpness of tuning between different cells or between different studies in that, unlike the preferred direction (μ), the individual κ values varied enormously, as much as 100-fold, depending on the parameter of the response that was being measured and the properties of the stimulus that evoked it. The κ values for the cells in our DSGC data set ($n=26$) ranged from 0.17 to 18.8 (mean 1.2 ± 0.37 std) when based on spike frequency and from 0.22 to 1.17 (mean 0.57 ± 0.12 std) when based on spike number. It was always the case that a response metric that had a threshold, such as spike generation, was more sharply tuned than a graded response, such as membrane potential or a dendritic Ca response. And in all cases tuning sharpness, but not direction orientation, was stimulus dependent. For example, the sharpness of directional tuning of spike responses decreases with increasing stimulus contrast. For these reasons we felt and continue to maintain that unlike the case of an electrical filter with fixed LRC components it is not useful to discuss the "bandwidth" of directional tuning of DS retinal cells without standardizing the response feature being measured and the stimulus that evoked it. In the absence of doing this it is misleading to represent the sharpness of directional tuning of the DS cells we have studied with a single number.

- Line 148. You don't know the "precise" location. Do you know the approximate location?

We removed the word "precise". This reference was meant to point out that we did not know the precise location of the fovea (in many cases fovea was not present in the retinal preparation that was recorded from and we could only estimate its location) which then would not allow us to locate the recorded cells relative to the fovea with precision, though we did know the retinal quadrant and approximate visual field location. Again, as we point out in the Discussion, this aspect of the DS cells functional architecture would require an additional detailed study well beyond the scope of this paper.

- Lines 200-212: This description of the circuitry is complex, and really taxes the reader. You help by referring to yellow balls and red balls in describing inputs onto dendrites. Continue this assistance, by adding mention of blue balls, green balls, and magenta balls where appropriate. Do I have it right: A1 amacrine cells get input from bipolar cells and project back onto bipolar cells?

Thank you, yes that is correct. We have included reference to the color notations related to the synapses now at all possible places in the text for further clarity.

Line 220: We identified starbursts displaced to the ganglion cell layer, as others have in non-primate mammals . . .” It would be nice to add to this paper reference to Yamada ES (2003). Also reference her paper with Marshak DW (2005) – she identified a potential direction selective ganglion cell. Did she get it right?

Our reference here was to the method of targeting these cells in vitro for physiological study (i.e., by the size and shape and density of cell bodies viewed in the living GCL with infrared viewing of tissue and recording pipette). In the 2003 paper by Yamada and Marshak noted by this reviewer the authors did not target starbursts for physiological study but simply used immunostaining of cholinergic processes in fixed tissue. In the second 2005 paper, some ganglion cell types are described qualitatively and mapped onto our Results from 2003. They make the suggestion, based on morphology that the Broad Thorny cell type may be direction selective but we and others have not found this in recordings from this cell type. They also suggested that another cell type might be the correlate of the ON-DS type described in rabbit retina. This cell resembles our recursive monostratified cell and as we suggest could represent the ON-DS type in primate but this remains to be shown directly. Yamada and Marshak (2005) is now cited along with other previous work attempting to characterize primate retinal ganglion cell types.

Line 359-360: the authors suggest that the DSGCs they have identified may account for the directional selectivity of cells in the accessory optic system. Why didn't they inject the nuclei comprising the accessory optic system, to test this idea directly? Or maybe they did label the AOS ganglion cell inputs when they made “pretectum” injections, but didn't realize it? Along these lines, what is meant by “pretectum”? Do you mean nuclei governing the pupil response (e.g., olivary nucleus)?

As stated above our “pretectal” injections were focused on the pretectal olivary nucleus (PON) and because of the proximity of NOT and DTN we assume that our injections encompassed all of these structures. We have now made this clear in the Methods section (L 457-464). Whether our suggested ON-DS types (the recursive monostratified cell) or the ON-OFF DS type (recursive bistratified cell) projects to AOS nuclei is an open and interesting question. The first step in addressing this question will be to determine the identity of the ON-DS cells in NHPs which would be the likely candidates for a projection to the AOS. In response to Reviewer 3 we have removed the text in the Discussion related to the AOS.

It is worth noting here that our “pretectal” tracer injections were directed at physiologically identified PON because at that time our focus was on identifying the melanopsin expressing ganglion cell type in the primate which we believed projected to the PON to drive the pupillary light reflex.

Line 425: The reader understands that the 29 animals used for rhodamine dextran injection were among the 76 used for this study. If that is not true, please clarify. Some readers may wonder why it was necessary to inject so many animals with rhodamine dextran in order to retrogradely label ON-OFF DSGCs. Perhaps a bit of explanation might be useful, such as “a typical

experiment allowed us to characterize the physiology of a single ON-OFF DSGCs; a total of 26 such neurons were studied in this report”.

Yes, it is true a subset of animals were used for the tracer injection experiments. The number of animals used reflects that we made injections into differing locations (pretectum, LGN and SC) in different sets of experiments and also addressed different physiological and anatomical questions over the course of these experiments and over many years. In addition, in some experiments results were minimal because either the tracer injections failed to some degree or there was some technical difficulty during retinal preparation that limited our ability to collect physiological data. We now show in the new Main Figure 2 that the data for the ON-OFF DSGCs was derived from 5 animals and the sample from each animal is color coded. We have also expanded on this aspect in the Methods section to make it clear to the reader (465-467).

FIGURES:

*In Fig. 1b, the ensemble of polygon sketches seems rotated clockwise with respect to the drawings of the actual cells. Could you orient them identically?** Where in the LGN was the injection made (parvo, magno?). What is the eccentricity of these cells?

This is now in Fig. 2a. Orientation has been fixed to align with the mosaic shown for this patch. Eccentricity is peripheral and is now noted in the figure caption. Injection was in the SC for this particular patch and is now noted in the caption as well.

**Moving on to Extended data figure 1, I now see that you often show a photomicrograph, drawing, and polygon sketch for cells, but each at a different orientation. Why not show each oriented consistently? It would be much easier for readers to make the transition between each representation of the data.*

In most cases in extended data Fig. 1 the polygons are not taken from the cells shown in that figure and are meant to show an example of the tiling/overlap for that cell type's mosaic. However, where we do show the same mosaic for both dendritic morphology and polygons (e.g., smooth monostratified cells) we have made sure everything is at the same orientation.

Fig. 1e, what does color signify? What does the single cell indicated by a star signify? The concentric rings in the plot represent the DSI, no? If so, why does it say “Spikes (#)” above the plot? Why the highly skewed distribution of favored directions for this sample of 26 cells? That's what you might expect if you kept hitting the same AOS subnucleus.

This plot is now shown in Fig. 3D and the colors have now been adjusted to indicate the sample from 5 different retinas. Yes, the extra Spike # tag was a mistake and has been removed. Injections were made into the LGN (both parvo and magno layers) or the SC. As noted in response to similar comment from Reviewer 1 the bias represents the limited sample and the fact that in a few retinas we recorded from patches of recursive cells at about the same locations. To reiterate – our hypothesis (that remains to be fully tested) is that there is a single

population of these ON-OFF DSGCs whose preferred direction will vary around the clock with retinal location in relationship to the fovea, as discussed in the Discussion section of the MS.

- Line 896: leave out “near complete”. There are 19 ganglion cells types. If mice have 40 types, this atlas is probably not “near complete”.

We actually do believe this is a “near complete” complete accounting of the ganglion cell types in the primate retina. The point of this figure is that by calculating densities from the mosaics and dendritic field overlap of each type we have accounted for over 95% of the total ganglion cells. It is possible that there are a few very low-density types that remain to be discovered. We argue therefore that there is a real species difference in the number of GCs in the primate retina and most of this may be due to the great increase in the number of DS types in mouse. We also note in the text that our data is consistent with recent data from the marmoset¹³ and also from recent transcriptomic profiling of primate neuronal types¹⁴.

- Line 898: “a small fraction of the total ganglion cells that project to both LGN and superior colliculus”. Do you mean that there is another cell type, besides recursive bistratified ganglion cells, that accounts for a much larger fraction of the cells that project to both targets? Do you mean sending a bifurcating axon to both targets?

We have rewritten this part of the caption to avoid confusion simply stating that this cell type by our analysis comprises a single type that constitutes 1.5% of the total ganglion cell population in the peripheral retina. As stated in response to a previous query we assume that this cell type projects to both structures by a branching axon as is the case for most other types.

- Line 913: those are not tracings of the cells shown in a, but rather the left panel of b. Can you detect any morphological difference between the parasol cells that project to SC versus LGN?

We fixed this by moving the lettering for each panel to its upper left-hand corner.

We did a study on the parasol cells that project to the SC¹⁰ and found no morphological differences. There are just two parasol cell types – ON and OFF/ inner and outer stratifying and they can be retrogradely labeled from tracer injections into either the LGN or SC, presumably like other “alpha-Y” type cells with large cell bodies that have a branching axon.

- Is the cell in the left panel of Extended Data Figure 1c among the cells drawn in the middle panel?

No, the cell in left panel is a separate isolated example. The cluster of 3 cells shown in the center are different cells. We now note this in the caption.

- Are the cells shown in Extended Data Figure 1f the same as those in Figure 1b, but now rotated and colored differently? If no other examples are available, and therefore you want to duplicate the illustration of these cells, please show them identically, don’t mess around with them.

Yes, this is the same patch of cells; we have now made sure the colors match but we did need to rotate the mosaic to fit into the panel configuration. We think this is OK, as the supplemental figures are in a separate PDF and stand apart from the main figures. However, we now also note the rotation (made for simplicity of presentation) in the Figure caption.

- *Can you explain the criteria used for the classification scheme presented in Extended Data Figure 1? Was it based purely on cell morphology, and dendrite stratification, relative to the ChAT bands in the IPL?*

It was based principally on the identification of cellular mosaics in which cell's dendritic trees are organized in relationship to their neighbors of the same type – from this data we can calculate the density of each type as a population and also consider other properties of that population that makes it distinctive (stratification depth, dendritic branching pattern, dendritic field diameter as a function of eccentricity, etc.).

- In Extended Data Figure 1, why does the distance between the lines representing the GCL and INL vary from panel to panel? Why does the width of the ChAT bands vary so much, and not always proportionately to the distance between the GCL and INL lines?

These properties are not meant to vary and some distortion may have occurred when we were resizing objects to create the figure. We have now checked this for uniformity and please note the Chat band depth, indicated in each panel is the same. The schematic representation is approximate but the numbers are meant to reflect the actual measurements.

- It seems like clutter to put 30 ± 6.9 and 63 ± 7.3 on every single panel.

Yes, we have removed this except for the first panel a and noted in the caption that the depth measurement applies to all panels in this figure

- Panel j, write “giant melanopsin cells” in place of “Giant MOPS”.
Thank you. Fixed.

- There seem to be 17 types of RGCs in Extended Data Figure 1, not 19.

Thank you for catching this. As noted in the new Main Fig. 1 and elaborated in Supplementary Figure 2 we have found evidence for 3 spatially overlapping mosaics of recursive monostratified cells. Although this is described in the caption we neglected to show in the right side of Supp. Fig. 1i. We have now fixed this to match what is shown in the new Figure 1. It is for this reason that the total is indeed 19 types.

- *Line 1030: change “extened” to “extended”*

This typo is no longer relevant as this Figure has now become Main Figure 1.

- *Extended Data Figure 2: What is the point of including this figure? Are “Recursive*

monostratied” (do you mean monostratified?) cells direction selective? Why are these cells getting special attention in this manuscript?

As noted above we found evidence that the recursive monostratified cells form 3 overlapping mosaics in which the dendrites fasciculate in bundles and this is what we show in Supplementary Figure 2. We felt this was important for several reasons. 1) It was part of our calculation for the total density of ganglion cells, 2) These cells are clearly candidates for the ON-DS type in the primate and we discuss this in the MS text and, 3) The obvious multiple overlapping and cofasciculating dendrites of these types (a known property of both ON and ON-OFF DS types with differing preferred directions in mouse and rabbit) contrasts sharply with what we found for the recursive bistratified cell where we only observed a single mosaic with a coverage near 1 (i.e., little overlap as shown in Fig. 2).

Extended Data Figure 4: I’m also puzzled why this figure is included. It is cited on line 125 to support the fact that a rare ganglion cell type showed recursively branching dendrites that bistratified in the IPL, but the figure doesn’t mention or show this rare ganglion cell type. Instead, it shows examples of parasol, midget, small bistratified, and broad thorny cells.

We cited this Figure (now Supplementary Fig. 3) to indicate that the recursive bistratified cells send dendrites into the Chat bands (shown schematically now in Main Fig. 2d). We used this figure to support our ability to measure dendritic stratification relative to the Chat bands in the primate. The Chat staining was combined with intracellular fills of the major cell types (parasol, midget and small bistratified) as well as the broad thorny (which is stratified broadly in the center of the IPL precisely in between the Chat bands). This allowed us to determine the relative stratification of all other cell types including the recursive bistratified cell. We have now tried to explain this more clearly in the Methods L-534-545).

- *Extended Data Figure 5: it is amazing to capture the transition from dendrite to axon. Superb!*
OK, thank you; this is now Supplementary Fig. 4.

- *Extended Data Figure 7: caption says the resting potential is ~75 mV, but it looks like -65 mV.*

This is now shown in Main Fig. 7a. We fixed this error to show the baseline potential around -75 mV.

- Why are the schematics of the receptive fields shown at different sizes for a, b, and c?
This is fixed. These panels are now in separate figures – Main Figure 7a and Supplementary Fig. 6.

- In (a) the maximum spot size listed is 720 um, but in the caption it says 800 um.
Thank you, fixed.

- Extended Data Figure 8: line 1132: legend seems garbled here.
This is now shown in Main Fig. 6e-g. The caption has now been clarified.

- Fig. 2: Are b and c showing the same field?

This is now Main Fig. 4b-c. Yes, and this has been clarified in the caption.

- *What should the reader make of the difference in m (180 degrees) and n (130 degrees) in optimal direction for the slit? Are the dendrites and axons tuned to different directions? Would different dendrites in the same cell have the same preferred direction?*

All of the above are possible. We now know that these cells are DS but we do not understand how DS is represented on different dendrites or different axons on the same cell. This is a major new question concerning the organization of DS in the mammalian retina. We consider this question briefly in the Discussion (L 414-416).

- *Fig 3: There seem to be many synapses visible in the EM (b) that are not shown in the reconstruction (h). Is that correct?*

This is now Main Fig. 5. Yes, b is a single section, many synapses are present in this section. In the reconstruction we are showing only the bipolar cell ribbon contacts (red balls) and for simplicity (because the density of inhibitory inputs from other amacrine cells is so high) we only showed both the inhibitory and excitatory synapses made on the A1 cell for the dendritic segment between the black arrows in h (red and yellow balls)

- *Is (h) simply intended to provide examples of various types of synapses, or is it supposed to show all that are visible in the EM (b)?*

We show all of the bipolar synapses present on this piece of the dendritic tree (red balls) – which target the dendritic spines. For the inhibitory synapses, because there were so many of them we only show the distribution on one length of dendrite as an example (yellow balls between the arrows indicated in (h)). We hope this is now clear in the Figure caption for Fig. 5.

- *How are contacts with bipolar cells identified with surety? By relying on ribbon morphology?*

Yes, the bipolar cells make ribbon synapses onto the A1 cell spines; an example is shown in (d) at a zoomed view. All bipolar synapses (whether on the starburst or the A1 cell or on midget and parasol ganglion cells) were determined by reconstructing the bipolar axon terminals and localizing all ribbons in the bipolar cell axon terminals that contacted a given dendrite, an extremely labor-intensive task to say the least.

Reviewer #3 (Remarks to the Author):

1. Kim and colleagues provide data of outstanding importance to understanding sensory processing in non-human primate (macaque monkey) retina. The data are also of high relevance to understanding human vision because macaques are evolutionary much more closely related to humans than the (nowadays) most common mammalian model system for visual processing i.e. mice. The brevity of this review does not reflect a lack of enthusiasm for the main result, which is that primate retina houses circuitry for direction selectivity (DS) largely aligned to that of better-studied mammals (mice, rabbits). In a trivial sense the broad result is confirmatory but at a

deeper level it is very important because the presence of direction-selective circuitry in primate retina has been repeatedly called into question as part of the "two-or-maybe-three afferent channel" view that still dominates in textbooks. Overall the authors deserve high praise for this study, which in my opinion deserves rapid publication without too much nit-picking by DS cognoscenti.

2. The paper is for the most part very clearly written and the results are compellingly documented and illustrated. I did have trouble to connect the dots at some places. These are detailed below. My only major suggestion for improvement is to ask whether the authors really want to present the (stunningly documented) survey of non-DS cell types here, or rather publish it elsewhere. It will be buried alive in this paper, which is about DS, yet deserves to be accessed by a broad vision science readership. There is increasing interest, for example, from psychophysics modeling and retina prosthesis communities on primate retinal cell populations, but those potential readers will very unlikely stumble upon the valuable data presented here as extended figure 1. (The fact that the legend to extended figure 1 runs over five manuscript pages should I think give the authors pause for thought).

As we outlined above we have considered all three reviewers comments related to the previous "extended data" figures and have moved a large fraction of this material to the main figures. With regard to the GC types component we do appreciate the reviewer's sentiment but on balance we believe that this platform is the best place for the non-DS types. One reason was that our major question was whether we were missing additional ON-OFF DS or other types so we took the time to acquire the mosaic data and to estimate the relative densities of all of our known types – and this data is presented for the first time here and supports the conclusion that there is only a single ON-OFF DS type in the primate retina. We have tried to draw attention to this material (so that it will not be "buried alive") by now beginning the paper with our anatomical classification and relative densities (Fig. 1) and the anatomical data for the recursive bistratified cell (Fig. 2). This places the density and cell types data up front and hopefully will be the basis for the interested reader to locate the additional details in the supplementary figures. It also allowed us to shorten the figure caption related to the GC cell types in the supplemental section considerably.

Minor:

- 3. I am not a member of the starburst fraternity therefore my comments to the starburst physiology do not come from a place of deep expertise. But even for a non-specialist there are some impediments to understanding. Starting from line 39 "generation of radial motion sensitivity ..." and popping up in many other places are strange circumlocutions and qualifications, where the authors seem to have doubts what relevance radial motion selectivity has for starburst responses to real-world stimuli. These are doubts which I share. The expanding ring stimulus does a brilliant diagnosis of centrifugal DS in starburst dendrites but such stimuli do not correspond to contours or movement in the external environment. Statements such as (l 399) "this excitatory surround is a critical element ... radial gratings" and elsewhere kept tripping me up: do the authors think that their conclusions about the bipolar-mediated surround apply to real-world stimuli or not? Some more clarity would be appreciated.

See response below with related points 13 and 14

4. L76 / *In the primate / In the primate retina /*
Thank you, fixed.

5. L82-94 "*distinctive ... unique ... Unexpectedly*" [*suggest remove or temper advertorials*]
Thank you, fixed

6. L91 "*distinct antagonistic ...*" *I may be wrong, but at places it seems the authors want to emphasise differences between monkey and mouse / rabbit, but don't the similarities rather outweigh the differences? Mouse/rabbit starbursts also have a strong antagonistic surround to my understanding, and the evidence the authors provide that GABA is not involved in monkey is not terribly strong. They do show that a GABAa antagonist has negligible effect but that is only a single null result and they do not provide a positive control. Many readers would appreciate more balanced claims and discussion of limitations.*

The reviewer is correct in "understanding" that starburst cells are known to have an antagonistic center-surround receptive field organization. The sentence queried by the reviewer (L91) was not meant to claim that the discovery of an antagonistic surround in primate starburst cells was unexpected. It was meant to say that we were surprised (did not expect) to find that surround "inhibition" in primate starburst cell arose presynaptically by modulation of an excitatory input rather than postsynaptically from increased inhibitory input as is thought to be the case in rabbit starburst cells¹⁵. We thank the reviewer for bringing this to our attention. We have clarified the intent of the sentence by removing our subjective response to the discovery of the presynaptic origin of starburst surround inhibition.

We do not agree with the reviewer that the evidence that surround inhibition is not mediated by GABAergic postsynaptic inhibition is "not terribly strong". We show that illumination of the surround generates a strong excitatory OFF response (Figure 6d, Supplemental Figure 6a) that could be generated at light OFFset by either disinhibition of inhibitory synaptic input or excitation (activation) of excitatory synaptic input. The voltage clamp recordings in supplemental figure 6b show that the OFFset of surround illumination evokes an excitatory (inward) synaptic current that decrease as the holding potential in moved closer to zero mV. This is evidence that the excitatory surround OFF response is not generated by disinhibition, which would get larger as the holding potential moved toward zero mV, i.e. moved in the positive direction from the Cl reversal potential. We maintain that voltage clamp recordings provide strong (convincing) evidence that the surround OFF response is generated by excitatory synaptic input, not by elimination of inhibitory synaptic input.

Furthermore, the surround OFF response persists in the presence of GABAzine (SR95531), well-documented selective GABAa receptor antagonist. It is standard and commonly used pharmacological tool for the characterization of GABAergic circuits in the CNS and retina. That it is an effective blocker of GABAergic inhibition has been demonstrated in numerous retina studies over the past 30 years including many in our lab^{16,17}. We feel these studies provide an

adequate "positive control" for the effects of GABAzine on inhibitory synaptic transmission in the retina.

7. L102-104 *I am sorry to be rude but this does seem a rather flimsy motivation for providing the encyclopedia of ganglion cells given in extended data figure 1. As noted above these data may deserve a better forum.*

We have now tried to strengthen this part of the text (L 101-108) and we hope that the changes we have made to the Figures as described above addresses this comment as well.

8.L107 "singular" [unclear]

Removed.

9. L107 "photodynamic" [elsewhere the terms "photo-stained" (and on line 982 "photostained") are used, it perhaps it should be introduced here.

Fixed. photostaining or photostained used throughout.

10. L205 / *By contrast, the A1 axons /*

Fixed, thank you.

11. L212 *What is the conclusion/functional implication of these findings?*

We consider the question of the significance of the morphological polarization and the spiking output of these cells in the Discussion L414-416.

12. L233, L236 *"surprisingly", "striking" [unclear; apart from the null gabazine result these data are quite compatible with mouse/monkey, aren't they?]*

The reviewer is correct that the preference of the somatic voltage response for outward over inward motion of a bullseye grating shown in figure 6c is compatible with data from rabbit and mouse starburst cells. What is surprising is the magnitude of the difference. In the studies that used a bullseye stimulus to document the starburst cell's directional preference the difference in the somatic voltage responses to outward versus inward motion was less than 3 mV (see: Euler et al 2002 (Fig. 7)¹⁸; Hausselt et al 2007 (Fig.1)¹⁹; Ezra-Tsur et al 2021 (Fig. 1i)²⁰). It was thus unexpected that the difference in the somatic voltage responses to outward versus inward motion in primate starbursts were typically greater than 20 mV, nearly 10 time larger than the difference in rabbit and mouse starburst cells. Since only the very few (if any) readers that were keenly aware of these differences in response amplitudes would find this "surprising" we have revised the sentence. It now reads:

"In macaque, the somatic voltage response to such a 'bullseye grating' showed a large unequivocal preference for outward motion (Fig 6c)."

- 13. L328-345 *This passage is quite impenetrable, it is not clear how the distinct temporal properties and/or spatial layout of bipolar inputs influenced the model results. The extended F11 does not show any parametric variation but rather two examples.*

- 14. Line 1208 *Surround delay 40 ms. This seems inconsistent with in vivo recordings e.g Smith et al., J Physiol 1992 found 5-10 ms surround delay.*

- 17. L1204 -1206 *this sentence is very unclear.*

Here we respond to points 13, 14 and 17 together since they are all related to our modeling exercise. We regret that the presentation of the model results was at times confusing and agree now that we could have done this better. Our goal with the model in this study was to test whether our hypotheses about the mechanisms underlying some physiological observations made with starbursts were well-founded. The three observations were (1) loss of the starburst surround following application of HEPES, (2) reduction of a directional preference in starbursts for centrifugal *versus* centripetal motion for the expanding/contracting radial grating stimulus following application of HEPES, and (3) the relationship between second harmonic and spatial frequency for counter-phase square-wave stimuli. Reviewer #3 drew attention to two aspects of the model that seemed to be explained inadequately: (1) the unexpectedly long delay (40 ms) between center and surround receptive field components that we used, and (2) the different spatial distributions of midget (presumed relatively sustained responses) and DB4/5 (presumed comparatively transient responses) bipolar cells to the dendritic tree of the starburst. We presume that the differential distribution of midget and DB4/5 bipolars is important for starburst function, but it is not necessary to explain the physiological results we sought to model. The differential distribution of midget and DB4/5 bipolar cells was incorporated to match the connectomic data on starbursts only. At the time of its incorporation we were unaware that it would be unnecessary for the observations we sought to model. The delay in the surround signal alone can account for the directional preference observed with the radial grating stimulus. Removing the surround, as is known to occur with HEPES, removes the directional preference. The effect of the surround delay on directionality is due purely to engagement of the midget bipolar cells, so the different spatial distributions of sustained and transient bipolar cell signals to the starburst dendritic tree, while surely of functional significance is not important in explaining the HEPES effect on directional signaling. The 40 ms delay for the surround signal did seem long to us also but was necessary to match physiological observations. The 40ms figure was derived from experiments that recorded the somatic voltage response to a spot of light, in which the application of HEPES eliminated a hyperpolarizing drop in V_m starting ~40-50ms after the initial depolarization. Additional simulations have been performed to show that the directional preference is not peculiar to a 40 ms surround delay (noted on lines 365-368 of the revised manuscript).

We agree with reviewer #3 that the radial grating stimulus is well suited to evoke directional signals from the starburst but unlikely to be commonly encountered in nature. Our reason for using that stimulus was to investigate the similarity between starburst properties in primate and those of mouse and rabbit. The radial grating has become a signature stimulus for studies of starburst physiology, so it was important for us to use it to compare the behavior of primate starbursts to that of starbursts of other species.

Interestingly, a recent bioRxiv pre-print explains how the delayed bipolar surround can affect responses to real-world stimuli²¹, and this reference is cited in the caption for Supplementary Figure 8. As noted above, we have revised the manuscript (in the modeling sections of Results and Methods, and in the caption for Supplementary Figure 8) to clarify the explanation of how the delayed surround of the bipolar cell comprises a mechanism for motion sensitivity with

radial stimuli. We have added in the Results modeling section a description of how different bipolar surround delays and stimulus velocities affect the directional differences in the starburst dendrites. Since the DB4/5 inputs to the starburst are peripheral, a delayed surround cannot generate a directional difference in their response to radial moving stimuli as it does for the central bipolar cells.

- Mention is made around line 336 of model responses to bars and gratings but data are not shown.

This data was placed in the extended data figure set in the original MS but has now been moved to the main figure set and appears in main Fig. 6e-g.

15. L358 My recollection is that Hoffmann's group interpreted OKN as more retinal-driven but pursuit as cortical-driven, that would be worth following up. In any event this sentence is rather simplistic.

We agree with the reviewer that this is a complex issue that we cannot address adequately in this Discussion. Because there is no evidence yet that ON-OFF DS cells even project to the NOT/DTN we removed this sentence from the MS.

16. L1039 this gallery does not include a recursive bistratified cell.

Our goal with this supplemental figure was to show how we measured the depth of the Chat bands in relation to various "marker" ganglion cell types that we intracellularly filled before Chat immunostaining. In the case of the recursive bistratified cells we determined stratification relative to the Chat bands by measuring their stratification relative to overlapping parasol ganglion cell dendritic trees at the same retinal location. This is now clarified in the Methods section (L 534-543).

References cited in this response to the Reviewers

- 1 Sabbah, S. *et al.* A retinal code for motion along the gravitational and body axes. *Nature* **546**, 492-497, doi:10.1038/nature22818 (2017).
- 2 Rodieck, R. W. & Marshak, D. W. Spatial density and distribution of choline acetyltransferase immunoreactive cells in human, macaque, and baboon retinas. *J.Comp.Neurol.* **321**, 46-64 (1992).
- 3 Huang, X., Rangel, M., Briggman, K. L. & Wei, W. Neural mechanisms of contextual modulation in the retinal direction selective circuit. *Nature communications* **10**, 2431, doi:10.1038/s41467-019-10268-z (2019).

- 4 Livingstone, M. S. Mechanisms of direction selectivity in macaque V1. *Neuron* **20**, 509-526, doi:10.1016/s0896-6273(00)80991-5 (1998).
- 5 Rossi, L. F., Harris, K. D. & Carandini, M. Spatial connectivity matches direction selectivity in visual cortex. *Nature* **588**, 648-652, doi:10.1038/s41586-020-2894-4 (2020).
- 6 Movshon, J. A. & Newsome, W. T. Visual response properties of striate cortical neurons projecting to area MT in macaque monkeys. *J. Neurosci.* **16**, 7733-7741 (1996).
- 7 Hawken, M. J. *et al.* Functional Clusters of Neurons in Layer 6 of Macaque V1. *J Neurosci* **40**, 2445-2457, doi:10.1523/JNEUROSCI.1394-19.2020 (2020).
- 8 Dacey, D. M., Peterson, B. B., Robinson, F. R. & Gamlin, P. D. Fireworks in the primate retina: in vitro photodynamics reveals diverse LGN-projecting ganglion cell types. *Neuron* **37**, 15-27 (2003).
- 9 Dacey, D. M. *et al.* Melanopsin-expressing ganglion cells in primate retina signal colour and irradiance and project to the LGN. *Nature* **433**, 749-754 (2005).
- 10 Crook, J. D. *et al.* Y-cell receptive field and collicular projection of parasol ganglion cells in macaque monkey retina. *J Neurosci* **28**, 11277-11291, doi:10.1523/JNEUROSCI.2982-08.2008 (2008).
- 11 Yang, G. & Masland, R. H. Receptive fields and dendritic structure of directionally selective retinal ganglion cells. *J. Neurosci.* **14**, 5267-5280 (1994).
- 12 Puller, C., Manookin, M. B., Neitz, J., Rieke, F. & Neitz, M. Broad thorny ganglion cells: a candidate for visual pursuit error signaling in the primate retina. *J Neurosci* **35**, 5397-5408, doi:10.1523/JNEUROSCI.4369-14.2015 (2015).
- 13 Masri, R. A., Percival, K. A., Koizumi, A., Martin, P. R. & Grunert, U. Survey of retinal ganglion cell morphology in marmoset. *J Comp Neurol* **527**, 236-258, doi:10.1002/cne.24157 (2019).
- 14 Peng, Y. R. *et al.* Molecular Classification and Comparative Taxonomics of Foveal and Peripheral Cells in Primate Retina. *Cell*, doi:10.1016/j.cell.2019.01.004 (2019).
- 15 Taylor, W. R. & Wassle, H. Receptive field properties of starburst cholinergic amacrine cells in the rabbit retina. *The European journal of neuroscience* **7**, 2308-2321, doi:10.1111/j.1460-9568.1995.tb00652.x (1995).

- 16 Crook, J. D., Manookin, M. B., Packer, O. S. & Dacey, D. M. Horizontal cell feedback without cone type-selective inhibition mediates "red-green" color opponency in midget ganglion cells of the primate retina. *J Neurosci* **31**, 1762-1772, doi:10.1523/JNEUROSCI.4385-10.2011 (2011).
- 17 Crook, J. D., Packer, O. S. & Dacey, D. M. A synaptic signature for ON- and OFF-center parasol ganglion cells of the primate retina. *Visual neuroscience* **31**, 57-84, doi:10.1017/S0952523813000461 (2014).
- 18 Euler, T., Detwiler, P. B. & Denk, W. Directionally selective calcium signals in dendrites of starburst amacrine cells. *Nature* **418**, 845-852 (2002).
- 19 Hausselet, S. E., Euler, T., Detwiler, P. B. & Denk, W. A dendrite-autonomous mechanism for direction selectivity in retinal starburst amacrine cells. *PLoS biology* **5**, e185, doi:10.1371/journal.pbio.0050185 (2007).
- 20 Ezra-Tsur, E. *et al.* Realistic retinal modeling unravels the differential role of excitation and inhibition to starburst amacrine cells in direction selectivity. *PLoS computational biology* **17**, e1009754, doi:10.1371/journal.pcbi.1009754 (2021).
- 21 Strauss, S. *et al.* Center-surround interactions underlie bipolar cell motion sensing in the mouse retina. *bioRxiv (Preprint)*, 1-30 (2021).

REVIEWER COMMENTS

Reviewer #2 (Remarks to the Author):

The authors have replied with great care and attentiveness to the extensive comments provided by each of the 3 reviewers. I like the way they have reorganized the paper and decreased the number of supplemental figures. Starting the paper with the atlas of identified cell types is a good idea, and the order of subsequent figures flows more logically.

I accept the authors' statement that identifying the target (s) of direction selective RGCs would require more restricted tracer injections, and is beyond the scope of this report. Lines 477-478 are garbled but seem to say that the data in the paper about recursive bistratified DSGCs are based on 5 retinas. Are the injection sites for these specific 5 retinas previously published? If so, it would be useful to give the precise reference and figure number (in their rebuttal, the authors state that they have previously published images of LGN and SC injection sites, but don't make it clear if they are referring to the injections that filled these 5 retinas). On line 130 it is stated that the cells were filled by tracer injection into either the LGN or the SC. In the methods section it is stated that PON injections might have included the NOT and DTN. I think that injections into the SC might also have deposited tracer into the NOT and DTN. I leave it up to the authors how to handle this – they seem very confident that the direction selective RGCs project to both the LGN and SC, but I'm not entirely persuaded without more evidence.

I could not understand the authors' response to my request for data about the bandwidth of direction tuning. Their response was very long and sophisticated, but not helpful. There should be some way to give an assessment of bandwidth or circular variance, even if an imperfect measure. Take the cell in Fig. 3c, for example. It is very directional, but also has quite a broad tuning in the favored direction.

Overall, this paper will constitute a valuable contribution to the literature and reflects years of careful and innovative studies by one of the top retina labs in the world.

Reviewer #3 (Remarks to the Author):

1. The authors have made a very nice set of revisions. I have only minor residual points.

2. It's a moot point, but in their response document the authors cite their previous studies as positive controls for effectiveness of Gabaa receptor blockade agents in monkey retina. But Crook et al., (2011, p 1767) stated "We found no effect [of Gabazine] on the spatial tuning functions or response phase ... ", and Crook et al., 2014, p65) state "GABAC receptor antagonists ... elicited small and variable changes ... ". It is the nature of these result (i.e. null results) which makes caution advisable. In other words, the same results could have been obtained by administration of an inert or badly degraded agent. Apologies if I was not making myself clear.

3. 143 / A second advantage / [unclear antecedent]

4. 271 / with the GABAa ... with the addition ... / [unclear]

5. 434. it's the authors' choice, but it might be prudent to point out that the surround delay from your model is longer than implied from in vivo studies e.g. Smith et al.

6. 463 it might be useful to tell readers at this point (or in connection with figure 3) that the preparation procedure did not allow precise identification of the position of recorded cells relative to the fovea. Some readers will be left wondering why nearly all points in Fig. 3d are above the horizon. Apologies if this information has been tucked away somewhere in the revised manuscript.

7. The author response document indicates that Figure 6 e-g show model responses to bars and gratings but Figure 6 shows _real_ cell responses, no? Line 1302 mentions model responses to linear bars and gratings but no data is shown. It's not a big deal, but perhaps at 1304 / sensitivity to motion / sensitivity to motion (data not shown) / would clear things up.

8. 463 / temperature was /

We again thank the reviewers for the precious time they put in on this detailed study, their helpful insights directed at improving our paper as well as their efficient response that helps publication in a timely manner. Their remaining comments and our response (in blue) are below with the line numbers in the manuscript noted where changes were made to the latest resubmitted MS version.

REVIEWER COMMENTS

Reviewer #2 (Remarks to the Author):

The authors have replied with great care and attentiveness to the extensive comments provided by each of the 3 reviewers. I like the way they have reorganized the paper and decreased the number of supplemental figures. Starting the paper with the atlas of identified cell types is a good idea, and the order of subsequent figures flows more logically.

I accept the authors' statement that identifying the target (s) of direction selective RGCs would require more restricted tracer injections, and is beyond the scope of this report. Lines 477-478 are garbled but seem to say that the data in the paper about recursive bistratified DSGCs are based on 5 retinas. Are the injection sites for these specific 5 retinas previously published? If so, it would be useful to give the precise reference and figure number (in their rebuttal, the authors state that they have previously published images of LGN and SC injection sites, but don't make it clear if they are referring to the injections that filled these 5 retinas). On line 130 it is stated that the cells were filled by tracer injection into either the LGN or the SC. In the methods section it is stated that PON injections might have included the NOT and DTN. I think that injections into the SC might also have deposited tracer into the NOT and DTN. I leave it up to the authors how to handle this – they seem very confident that the direction selective RGCs project to both the LGN and SC, but I'm not entirely persuaded without more evidence.

Tracer injections for retrograde labeling were made by physiologically mapping the structure (either the LGN or SC) by its response to a light pulse. Our first goal was to localize the rostral, caudal, medial and lateral borders and stay well within this zone. In the case of the SC, we made our injections into the visually responsive superficial layers (and our histology confirmed this – with an example shown in Crook et al., 2008). So, it is unlikely that these injections extended into the NOT/DTN. The physiological data were derived from 5 retinas (SC injections) but of course the anatomical data (122 cells, Figure, 2c) were derived from many retinas and included LGN and SC injections. Since virtually all ganglion cell types project to the SC except for the midget ganglion cells it is certainly possible that the recursive cells project to the NOT/DTN as well as the SC and just did not observe it in our “pretectal PON” injections. It is worth emphasizing that our goal was not a focused study on the central projections of the ON-OFF recursive bistratified cell but to demonstrate its direction selectivity for the first time. Certainly, we can conclude with some confidence that these cells project to the LGN and SC (Supplementary Table 1) whether they might also project to additional targets awaits a more refined approach to establishing the visual function of this GC type. In the Results and Methods sections, we have now made additional note of this caveat (Lines 132-136 and Lines 485-486).

I could not understand the authors' response to my request for data about the bandwidth of direction tuning. Their response was very long and sophisticated, but not helpful. There should be some way to give an assessment of bandwidth or circular variance, even if an imperfect measure. Take the cell in Fig. 3c, for example. It is very directional, but also has quite a broad tuning in the favored direction.

We understand and share the reviewer's interest in the bandwidth (BW) of direction tuning and we are sorry the reviewer did not understand the reasons we are not able to provide useful data about it at this time. The straightforward reason for deciding not to do this is that tuning BW, assessed either by visual inspection of polar plots or by analysis using directional statistics (von Mises fits), varied enormously depending on the parameter of the response that was being measured (i.e., spike number/frequency or response voltage peak amplitude/area) and the properties of the stimulus that evoked it (i.e., moving bar dimensions, velocity, and contrast). In view of this, we decided that a full description of the stimulus dependence of tuning BW was beyond the scope of our initial report of the origins of DS in the primate retina and thus warranted further study in a separate project designed specifically to address it. In this regard, it is also relevant to point out that we know of no study of retinal direction-selective that has meaningfully discussed, let alone quantified, the BW of directional tuning. And unless the reviewer knows of such a study, we are consequently being asked to provide novel information about an important parameter of DS, which makes it all the more important that it be documented in detail. However, we have now made this point more explicit in the Results section:

“We also note that the sharpness of DS tuning but not the preferred direction varied with the parameter of the response that was being measured (spike rate vs total spike count) and the properties of the stimulus (e.g., velocity and contrast) that evoked it but did not explore these parameter spaces in detail here.” (Lines 159-162)

Overall, this paper will constitute a valuable contribution to the literature and reflects years of careful and innovative studies by one of the top retina labs in the world.

Reviewer #3 (Remarks to the Author):

1. The authors have made a very nice set of revisions. I have only minor residual points.

Thank you!

2. It's a moot point, but in their response document the authors cite their previous studies as positive controls for effectiveness of Gabaa receptor blockade agents in monkey retina. But Crook et al., (2011, p 1767) stated "We found no effect [of Gabazine] on the spatial tuning

functions or response phase ... ", and Crook et al., 2014, p65) state "GABAC receptor antagonists ... elicited small and variable changes ... ". It is the nature of these result (i.e. null results) which makes caution advisable. In other words, the same results could have been obtained by administration of an inert or badly degraded agent. Apologies if I was not making myself clear.

Thank you for this additional comment and no, the initial comment about the lack of Gabazine effect was made clearly. A couple of additional points might assuage this reviewer's concerns. First, we can assure the reviewer that we purchase Gabazine quite regularly and have observed its effects (using 1, 5 and 10 micromolar concentrations) on ganglion cells consistently, typically elevating the spike rate and in the case of voltage clamp, clearly shifting the reversal potential to more positive values. The fact that HEPES and not GABAzine attenuates the surrounds of light adapted ganglion cells was strong evidence in support of a other results by multiple investigators leading to the conclusion that the negative feedback from horizontal cells to cones is, in fact, not mediated by GABAergic inhibitory synapse, and certainly no actual synapse from horizontal cells to cone pedicles has ever been found. So our result that HEPES, but not GABAzine, abolishes the surround response of starburst cells, is more evidence quite consistent with previous findings from our lab as well as others. An indication of starburst-to-starburst inhibition in the starburst light response and its role in starburst DS, if any, remains to be determined.

3. 143 / A second advantage / [unclear antecedent]
Fixed

4. 271 / with the GABAa ... with the addition ... / [unclear]
Fixed

5. 434. *it's the authors' choice, but it might be prudent to point out that the surround delay from your model is longer than implied from in vivo studies e.g. Smith et al.*

As we went through in our original response, we chose the surround delay in the model based on the surround latencies we recorded from the starbursts:

"The 40 ms delay for the surround signal did seem long to us also but was necessary to match physiological observations. The 40ms figure was derived from experiments that recorded the somatic voltage response to a spot of light, in which the application of HEPES eliminated a hyperpolarizing drop in V_m starting ~40-50 ms after the initial depolarization. Additional simulations have been performed to show that the directional preference is not peculiar to a 40 ms surround delay." (From the original response).

We emphasize this point further in the current text noting that there is an inverse relationship between stimulus velocity and surround delay: shorter delays, 10, 20 ms give a smaller but

similar qualitative result and that can be offset by increasing stimulus velocity. We have edited the text (Lines 375-380) to better make this point:

“In this regard it is worth noting the critical interaction between surround delay and stimulus velocity. Thus, models with a smaller surround delay (e.g., 20 ms vs. 40 ms) produced smaller but qualitatively similar outward directional preference in the starburst dendrites, that could be offset by models run with a higher velocity grating (e.g., 400 $\mu\text{m/s}$ vs. 200 $\mu\text{m/s}$) which produced a larger outward directional preference.” (Lines 375-380)

And in the Methods, we now cite Smith et al., as an example:

“While we recognize shorter surround delays in some previous models of primate ganglion cell receptive fields^{e.g.114} the modeled surround delay was derived from measurements of HEPES sensitive hyperpolarization response latencies in macaque starbursts.” (Lines 742-745)

6. 463 it might be useful to tell readers at this point (or in connection with figure 3) that the preparation procedure did not allow precise identification of the position of recorded cells relative to the fovea. Some readers will be left wondering why nearly all points in Fig. 3d are above the horizon. Apologies if this information has been tucked away somewhere in the revised manuscript.

Yes, we have made this point in the MS. The major reason for the apparent asymmetry is that many of the cells were sampled from the same location (same mosaic patch of cells) in a few retinas. The fact that foveal locations were roughly estimated in these early experiments may also have contributed. In the Results we note:

“The preferred direction was variable in this relatively small sample and foveal location was only roughly estimated, however cells recorded in nearby locations from the same retina tended to show very similar direction preferences (Fig. 3d).” (Lines 157-158)

7. The author response document indicates that Figure 6 e-g show model responses to bars and gratings but Figure 6 shows _real_ cell responses, no? Line 1302 mentions model responses to linear bars and gratings but no data is shown. It's not a big deal, but perhaps at 1304 / sensitivity to motion / sensitivity to motion (data not shown) / would clear things up.

Yes, we now note data “not shown” (Supplementary information, Line 259)

8. 463 / temperature was /

Yes, we fixed.